# Causal Explanation-Guided Learning for Organ Allocation

**Alessandro Marchese**[*]
Vrije Universiteit Brussel

**Jeroen Berrevoets**
King's College London

**Sam Verboven**
Vrije Universiteit Brussel

## Abstract

A central challenge in organ transplantation is the extremely low acceptance rate of donor organ offers—typically in the single digits—leading to high discard rates and suboptimal use of available grafts. Current acceptance models embedded in allocation systems are non-causal, trained on observational data, and fail to generalize to policy-relevant counterfactuals. This limits their reliability for both policy evaluation and simulator-based optimization. In this work, we reframe organ offer acceptance as a counterfactual prediction problem and propose a method to learn from routinely recorded—but often overlooked—refusal explanations. These refusal reasons act as direction-only counterfactual signals: for example, a refusal reason such as `"old donor age"` implies acceptance might have occurred had the donor been younger. We formalize this setting and introduce CLEXNET, a novel causal model that learns policy-invariant representations via balanced training and an explanation-guided augmentation loss. On both synthetic and semi-synthetic data, CLEXNET outperforms existing acceptance models in predictive performance, generalization, and calibration, offering a robust drop-in improvement for simulators and allocation policy evaluation. Beyond transplantation, our approach provides a general method for incorporating human direction-only explanations as a form of model supervision, improving performance in settings where only observational data is available.

## 1 Introduction

Organ transplantation is often the definitive treatment for end-stage organ failure. Yet demand for donor organs persistently outweighs supply [36]. As such, many patients deteriorate or die while waiting for a suitable donor [61, 28, 1]. Optimizing allocation systems to match donated organs with compatible recipients is therefore a critical task.

**Organ offer refusals cause widespread inefficiencies in transplant systems.** The acceptance rate of organ offers in the U.S. is extremely low: only 1% of kidney, 3% of liver, and 5% of lung offers are accepted [41]. Each refusal not only delays transplantation for the recipient but triggers a cascading effect across the system. Organs accumulate cold ischemic time as they are offered down the ranked waitlist, increasing the risk of graft failure and reducing transplant success [35, 38, 62, 20]. Moreover, extended offer chains burden allocation logistics and significantly increase the probability of eventual organ discard [42, 57].

**Existing acceptance models are oversimplified and unfit for counterfactual estimation.** Current ML-based allocation studies sidestep the complexity of refusals, assuming every offer is accepted [8, 6, 7, 65, 66]. This simplifying assumption breaks down in practice, leading to unrealistic simulations and misinformed policy guidance. Even real-world simulators used by transplant organizations that

---

[*]Corresponding author

39th Conference on Neural Information Processing Systems (NeurIPS 2025).

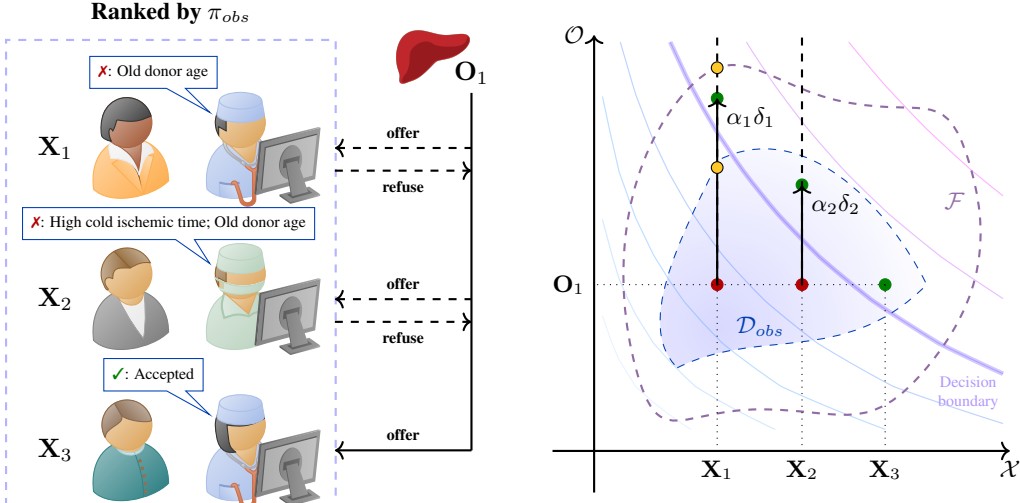

Figure 1: **Illustrative overview of organ offer acceptance. Left:** An incoming liver offer $O_1$ is broadcasted down the observed-policy $\pi_{obs}$ ranking. Each candidate $X_i$ either accepts (green tick) or refuses (red cross) and—if refusing—supplies a categorical refusal reason (call-outs). **Right:** Geometric view of the patient space $\mathcal{X}$ and organ offer space $\mathcal{O}$. The blue shaded region marks the domain covered by the observational dataset $\mathcal{D}_{obs}$; the dashed purple curve encloses the full feasible domain $\mathcal{F}$. Dashed arrows show atomic refusal-reason directions, solid arrows contrastive counterfactual edits $\alpha_i \delta_i$ that would cross the decision boundary and convert a refusal into an acceptance. For example, based on the factual samples in $\mathcal{D}_{obs}$ and direction $\delta_1$, a model should learn that the decision boundary lies between the intersected boundaries (the yellow points) of $\mathcal{D}_{obs}$ and $\mathcal{F}$.

incorporate acceptance models [49, 44, 45], fall short for two key reasons. First, they are typically based on logistic regression [54, 56, 55, 18, 19], which, while interpretable, lacks the capacity to model the complex, high-dimensional interactions involved in clinical decision making. Modern machine learning alternatives offer significantly greater expressiveness [64, 33, 10, 67, 40]. Second, these models are trained on observational data generated under the current policy, inheriting spurious correlations—manifestations of shortcut learning [23]—that lead to confounding bias [5], and failure to generalize to counterfactual policy-relevant scenarios. Even after adjusting for confounding bias, reliable generalization to unseen counterfactual policies remains difficult without additional sources of information.

**Refusal reasons are untapped direction-only contrastive explanations.** A key yet unused source of supervision to improve acceptance models lies in the refusal reasons recorded with each declined offer [12, 29]. These categorical reasons—for example, `old donor age` or `high cold ischemic time`—act as directional signals, suggesting how a donor or recipient attribute could be modified to change the acceptance outcome. Crucially, these signals are *contrastive* but not *quantitative*: they indicate which features to adjust, but not by how much, nor whether such adjustments are minimal, or whether the response surface is local or monotonic.

**Current explanation-guided methods cannot handle direction-only explanations.** Existing explanation-guided methods require richer forms of supervision such as precise counterfactual distances [24, 21], local input-gradient constraints [47, 59] or global monotonicity rules [25, 37]. As such, these methods are ill-suited for this setting in which only a sparse, directional signal is available. Bridging this methodological gap is crucial for building unbiased acceptance models and fully utilizing the explanatory power of this routinely collected data.

**Our contributions.** We propose a new learning framework that formalizes this novel supervision regime and introduce CLEXNET. Our contributions are threefold:

- We formalize a new learning setting in which the only extra supervision available is a set of categorical refusal reasons that point out a *direction*—but never the magnitude—in which donor or recipient attributes would need to change for an organ offer to be accepted.
- We construct a guarded *explanation-guided augmentation* scheme that converts each categorical refusal reason into a set of feasible counterfactual edits, enabling *any* classifier to learn from direction-only feedback without needing distance or monotonicity information.
- We introduce CLEXNET, a causally calibrated acceptance model that learns policy-invariant representations using adversarial balancing and an explanation-guided augmentation loss. This model simultaneously corrects for observational bias and respects directional refusal constraints.

Through comprehensive synthetic and semi-synthetic experiments, we demonstrate that CLEXNET outperforms existing acceptance models in generalization, calibration, and predictive accuracy—offering a practical and robust improvement for policy simulators. More broadly, our approach opens a new direction in counterfactual machine learning by operationalizing contrastive human feedback in high-stakes, observational settings like organ transplantation.

## 2 Problem formulation

During the organ offer process, the organ is repeatedly offered electronically to potential transplant recipients in a ranking until a potential recipient accepts the offer as shown in Figure 1. In this section, the offer made to each patient is modeled, and the mathematical notation is introduced.

**Making an offer.** Consider $\mathcal{X} \subset \mathbb{R}^{d_x}$ as the space of all possible patients and $\mathcal{O} \subset \mathbb{R}^{d_o}$ as the space of all possible organ offers. Let $\mathbf{X} \in \mathcal{X}$ and $\mathbf{O} \in \mathcal{O}$ be the feature vectors of a patient and an organ offer respectively.

After an offer has been made, an answer $Y \in \{0,1\}$, where 0 represents a refusal and 1 represents acceptance, is received from the patient. When a patient declines an offer, they must provide a reason. Consider $\mathcal{R} = \{r_1, r_2, \ldots, r_K\}$ as the set of all possible refusal reason categories. Each $r \in \mathcal{R}$ represents a discrete category for the reason why an offer might be refused. For example, these categories might correspond to issues related to the donor age or the cold ischemic time of the organ. Finally, let $R \in \mathcal{R} \cup \{\emptyset\}$ denote a variable representing such reason, allowing for the absence of a reason, represented by $R = \emptyset$, in the case of acceptance.

We assume we have an observational dataset containing $N$ instances of patient-offer pairs and their corresponding answer and reason, resulting in a dataset of observations of quadruplets $\mathcal{D} = \{(\mathbf{X}_i, \mathbf{O}_i, Y_i, R_i) : i = 1 \ldots K\}$, where all patients and organs are sampled from some underlying distributions $p(\mathbf{X})$ and $p(\mathbf{O})$. Finally, consider $\mathcal{Q} \in \mathcal{P}(\mathcal{X})$ [2] as a set that represents the wait list.

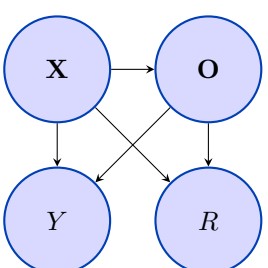

Figure 2: **Graphical structure of organ offer and patient response.** An organ offer $\mathbf{O}$ is extended, following the existing policy $\pi_{obs}$, to a candidate patient characterized by $\mathbf{X}$. The patient responds with a binary decision $Y$, which is determined by both the patient's features and the offer's features. In cases of refusal, the patient supplies a refusal reason $R$, which can be expressed in terms of $\mathbf{X}$ and $\mathbf{O}$ through a predefined mapping $\mathcal{M}$.

The dataset $\mathcal{D}$ is observed under policy $\pi_{obs} : \mathcal{Q} \times \mathcal{O} \to \mathfrak{S}_{\mathcal{Q}}$ where $\mathfrak{S}_{\mathcal{Q}}$ [3] denotes the generated ranking of patients in the wait list. The ranking $\mathfrak{S}_{\mathcal{Q}}$ directly dictates the sequence of potential transplant recipients are considered. As a result, the generation of the observed dataset $\mathcal{D}$ is inherently influenced by $\pi_{obs}$. A graphical representation of the underlying structure of $\mathcal{D}$ is shown in Figure 2.

For both the answers, $Y_i$ and the reasons $R_i$ we assume that they are generated following the Rubin-Neyman potential outcomes framework [48]. For each patient there are two sets of potential outcomes $\{Y(\mathbf{o}) : \mathbf{o} \in \mathcal{O}\}$ and $\{R(\mathbf{o}) : \mathbf{o} \in \mathcal{O}\}$, and that the observed outcomes $Y_i$ and $C_i$ are consistent with the potential outcomes $Y(\mathbf{O}_i)$ and $R(\mathbf{O}_i)$ for the observed offer.

---

[2]$\mathcal{P}(\cdot)$ is the power set of the given set.
[3]$\mathfrak{S}_{(\cdot)}$ is the symmetric group of a given set.

**Interpreting direction-only reasons.** Consider $\Delta = (\mathcal{X} \cup \mathcal{O}) \cap \{-1, 0, 1\}^{d_x + d_o}$ and let $\mathcal{M} : \mathcal{R} \to \Delta$ denote a function that maps refusal categories onto corresponding signed vector embeddings in the constrained patient-offer space $\Delta$. For any refusal category $r \in \mathcal{R}$, the embedded reason is $\delta \in \Delta$, given by $\mathcal{M}(r) = \delta$.

The embedded reason $\delta$ represents the direction of a contrastive counterfactual explanation [51]; in what direction the features of $\mathbf{X}$ and $\mathbf{O}$ have to change such that the offer would get accepted. As the complete counterfactual explanations are not disclosed, for each continuous feature we only encode one of three possible options for the direction: decrease (-1), remain unchanged (0) and increase (1).

**Assumption 1 (Counterfactual existence).** Consider $\delta_\mathbf{X}$ and $\delta_\mathbf{O}$ as vectors composed by the elements of $\delta$ that correspond to the features of $\mathbf{X}$ and $\mathbf{O}$ respectively such that $\delta = (\delta_\mathbf{X}, \delta_\mathbf{O})$. For any observed refusal with a refusal reason $R \neq \emptyset$ with corresponding embedding $\delta$, we assume that:

$$\exists \alpha_\mathbf{X} \in \mathbb{R}_+^{d_x}, \alpha_\mathbf{O} \in \mathbb{R}_+^{d_o} : Y(\mathbf{X} + \alpha_\mathbf{X}\delta_\mathbf{X}, \mathbf{O} + \alpha_\mathbf{O}\delta_\mathbf{O}) = 1, \tag{1}$$

where $\alpha_\mathbf{X}$ and $\alpha_\mathbf{O}$ are vectors that represent the needed magnitudes of change in the features of $\mathbf{X}$ and $\mathbf{O}$ respectively and $\alpha = (\alpha_\mathbf{X}, \alpha_\mathbf{O})$. In words, this assumption states that a positive counterfactual can be found if a factual negative sample is edited along the direction $\delta$.

**Assumption 2 (Feasible magnitude).** Let $\mathcal{F} \subset \mathcal{X} \cup \mathcal{O}$ be a realistic and feasible region of patient-organ pairs. Consider $\mathcal{A}$ as the space of all edit magnitudes such that:

$$\forall \alpha \in \mathcal{A} : (\mathbf{X} + \alpha_\mathbf{X}\delta_\mathbf{X}, \mathbf{O} + \alpha_\mathbf{O}\delta_\mathbf{O}) \in \mathcal{F}. \tag{2}$$

With this assumption, $\mathcal{F}$ limits the possible edit magnitudes such that the counterfactuals would remain in a specific, predefined region. For example, it would not make sense if $\mathbf{O} + \alpha_\mathbf{O}\delta_\mathbf{O}$ results in an organ from a donor with a negative age feature. Graphical representations of reasons transformed into counterfactual edits are shown in Figure 1.

**Causal assumptions.** To identify the effects of organ offers on patient responses we impose three standard causal assumptions. First, *positivity* (or *overlap*) requires that for every patient–offer feature pair $(\mathbf{X}, \mathbf{O})$ such that $\mathbf{X}$ has nonzero support in the wait-list distribution, the observed policy $\pi_{obs}$ assigns a strictly positive probability of that offer being made, i.e. $\Pr(\mathbf{O} \mid \mathbf{X}) > 0$. Second, *unconfoundedness* (or *ignorability*) assumes that, conditional on the patient and offer covariates, the pair of potential outcomes $\{Y(\mathbf{o}), R(\mathbf{o}) : \mathbf{o} \in \mathcal{O}\}$ is independent of the ranking and assignment mechanism, formally:

$$\{Y(\mathbf{o}), R(\mathbf{o})\}_{\mathbf{o} \in \mathcal{O}} \perp\!\!\!\perp \pi_{obs} \mid \mathbf{X}. \tag{3}$$

Finally, we assume the *stable unit treatment value assumption* (SUTVA), which consists of two parts: (i) *consistency*, meaning that the observed outcome and refusal reason coincide with the corresponding potential outcomes under the realized offer, $Y = Y(\mathbf{O})$ and $R = R(\mathbf{O})$, and (ii) *no interference*, meaning that one patient's response and reason are unaffected by the offers or decisions of any other patient.

## 3 CLEXNET

We introduce CLEXNET[4], a *causal–explanation–guided* acceptance model that jointly tackles three challenges:

(i) **Predictive accuracy** on the (biased) training distribution;

(ii) **Causal robustness**, i.e. invariance to the organ-allocation mechanism that generated the data; and

(iii) **Faithfulness to refusal reasons**, which convey only a *direction* in which features must change for an organ to be accepted.

To achieve these goals, the network couples an *adversarially–balanced representation* with a directional *explanation–guided augmentation loss*. Figure 3 gives a bird's-eye view of the CLEXNET architecture and Algorithm 1 details the training loop.

---

[4]CLEXNET components that correspond with our main contributions are marked with blue

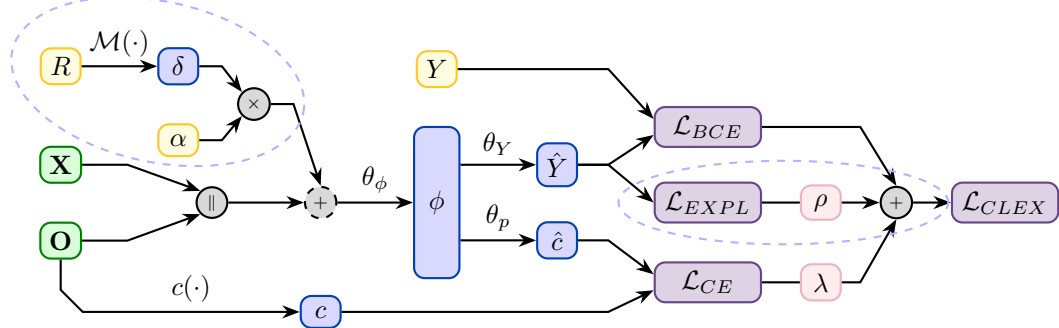

Figure 3: **Architecture of our CausaL, EXplanation guided organ-offer model (CLEXNET).**
Inputs $\mathbf{X}, \mathbf{O}$ go through a shared representation $\phi$, which feeds both a clustering head $c_{\theta_p}$ and an
acceptance head $Y_{\theta_Y}$. The observed refusal category $R$ is embedded as $\delta$ and combined with $\mathbf{X}$ and
$\mathbf{O}$ before passing it to $\Phi_{\theta_\phi}$. All three losses are aggregated into $\mathcal{L}_{CLEX}$. The plus operator with
the dashed outline indicates where generated counterfactual edits are injected during training. We
use blue to represent components related to representations, gray for simple vector operators, purple
for losses, pink for loss multipliers, yellow for components that are only present during training
and green for model inputs used at inference time. The components that correspond with our main
contributions are circled with dashed outlines.

**Balancing away confounding bias.** Under the observed allocation rule $\pi_{obs}$, patients receive offers
that are *not* independent of their covariates. Simply minimizing the binary-cross-entropy on such
data would entangle acceptance with the policy's selection shortcuts [52, 50]. In this setting, the
organs are sparse and complex high dimensional objects. Following previous work [8], we utilize
organ clusters $c_i(\mathbf{O}), i = 1 \dots k$ to reduce the dimensionality of the treatment space. These clusters
serve as surrogates for treatment in the balancing step. We aim at adversarially learning [22, 13, 8] an
intermediate representation $\phi$ which is invariant to the propensity of the cluster of the provided organ.

Consider an acceptance model $\text{CLEXNET}_{\theta_\phi, \theta_Y, \theta_p}$ with parameters $\theta_\phi$, $\theta_Y$ and $\theta_p$ such that the
balanced representation $\phi = \Phi_{\theta_\phi}(\mathbf{X}, \mathbf{O})$, the organ cluster $\hat{c} = c_{\theta_p}(\phi)$, and the acceptance probability
$\hat{Y} = Y_{\theta_Y}(\phi)$ are learned jointly.

Using these parameters, we construct a Cross Entropy (CE) loss component for the organ cluster and
a Binary Cross Entropy (BCE) loss for the offer acceptance probability:

$$\mathcal{L}_{CE}(\theta_\phi, \theta_p) := -\frac{1}{N} \sum_{i=1}^{N} \Big[ \mathbb{I}[\hat{c}_i = c(\mathbf{O}_i)] \cdot \log(\hat{c}_i) \Big], \tag{4}$$

$$\mathcal{L}_{BCE}(\theta_\phi, \theta_Y) := -\frac{1}{N} \sum_{i=1}^{N} \Big[ Y_i \cdot \log(\hat{Y}_i) + (1 - Y_i) \cdot \log(1 - \hat{Y}_i) \Big]. \tag{5}$$

We minimize BCE on the acceptance head *and* maximize cross-entropy error on the cluster head via
a gradient-reversal layer [22, 8]. Concretely, letting $\lambda \in [0, 1]$ control the trade-off:

$$\underbrace{\mathcal{L}_{BCE}(\theta_\phi, \theta_Y)}_{\text{Accuracy}} - \underbrace{\lambda \cdot \mathcal{L}_{CE}(\theta_\phi, \theta_p)}_{\phi \text{ uninformative about } c}, \tag{6}$$

where the multiplier $\lambda$ serves to control between balancing the representation $\phi$ and acceptance
prediction. Balancing guarantees that, conditioning on $\phi$, the empirical distribution of organ clusters
resembles a randomised trial, thereby attenuating selection bias [50, 13]. At this point, $\phi$ can be a
balanced representation, but the refusal explanations are not yet being used.

**Embedding reasons.** Besides predicting $Y$ and being balanced, we also aim for $\text{CLEXNET}_{\theta_\phi, \theta_Y, \theta_p}$
to respect the given refusal reasons. To be able to learn from the refusal reasons, they must first be
embedded in $\mathcal{X} \cup \mathcal{O}$. A fixed lookup $\mathcal{M}$ maps every reason to a signed vector and is used to transform
each observed quadruplet into $(\mathbf{X}_i, \mathbf{O}_i, Y_i, \mathcal{M}(r_i)) = (\mathbf{X}_i, \mathbf{O}_i, Y_i, \delta_i)$. Negative, zero, and positive
entries of $\delta_i$ respectively indicate whether to decrease, hold, or increase the associated feature.

**Algorithm 1:** CLEXNET : single–instance training step with explanation-guided augmented loss

---

**Input:** instance $(\mathbf{X}, \mathbf{O}, Y, R)$; clusters $c(\cdot)$; lookup $\mathcal{M}(\cdot)$;
domain $\mathcal{F}$; weights $\lambda, \rho$; edits $M$; threshold $p_{ex}$
**Init:** encoder $\Phi_{\theta_\phi}$; acceptance head $Y_{\theta_Y}$; cluster head $c_{\theta_p}$

$\phi \leftarrow \Phi_{\theta_\phi}(\mathbf{X}, \mathbf{O}); \hat{Y} \leftarrow Y_{\theta_Y}(\phi); \hat{c} \leftarrow c_{\theta_p}(\phi)$ ;        ▷ forward pass of factual
$\mathcal{L}_{BCE} \leftarrow \text{BCE}(Y, \hat{Y}); \mathcal{L}_{CE} \leftarrow \text{CE}(c_i, \hat{c})$ ;                  ▷ derive losses
$\mathcal{L}_{EXPL} \leftarrow 0$ ;                            ▷ initialize explanation loss
**if** $Y = 0$ **and** $R \neq \emptyset$ **then**
   $\delta \leftarrow \mathcal{M}(R)$ ;                               ▷ embedded reason
   **for** $m = 1$ **to** $M$ **do**
      $(\tilde{\mathbf{X}}^{(m)}, \tilde{\mathbf{O}}^{(m)}) \leftarrow \text{Sample}((\mathbf{X}, \mathbf{O}), \delta, \mathcal{F})$ ;     ▷ guided augmentation (Eq.1, 2)
      $p^{(m)} \leftarrow Y_{\theta_Y}(\Phi_{\theta_\phi}(\tilde{\mathbf{X}}^{(m)}, \tilde{\mathbf{O}}^{(m)}))$ ;         ▷ forward pass of counterfactual
   $p_{\max} \leftarrow \max_m p^{(m)}; p_{\text{avg}} \leftarrow \frac{1}{M} \sum_m p^{(m)}$ ;       ▷ max and avg of counterfactuals
   **if** $p_{\max} < p_{ex}$ **then**
      $\mathcal{L}_{EXPL} \leftarrow \text{BCE}(p_{\text{avg}}, p_{ex})$ ;         ▷ guarded explanation loss (Eq.7, 8)

$\mathcal{L}_{CLEX} \leftarrow \mathcal{L}_{BCE} - \lambda \mathcal{L}_{CE} + \rho \mathcal{L}_{EXPL}$;
$\{\theta_\phi, \theta_Y, \theta_p\} \leftarrow \text{Update}(\nabla \mathcal{L}_{CLEX})$ ;             ▷ optimization step (optional)

---

**Guarded explanation-guided augmentation.** Relying on Assumptions 1 and 2, we formulate an explanation loss component that is protected by a logical guard such that (i) the explanation loss is only considered for negative samples and (ii) the model does not already respect the explanation:

$$\mathcal{G}_i := \mathbb{I}[Y_i = 0] \cdot \underbrace{\mathbb{I}[\max_{\alpha_i \in \mathcal{A}_i} Y_{\theta_Y}(\Phi_{\theta_\phi}(\mathbf{X}_i + \alpha_{\mathbf{X}_i}\delta_{\mathbf{X}_i}, \mathbf{O}_i + \alpha_{\mathbf{O}_i}\delta_{\mathbf{O}_i}) \leq p_{ex})]}_{\text{CLEXNET's Assumption 1}}, \tag{7}$$

where $p_{ex}$ acts as a threshold that should be met by at least one augmented sample, offering a tunable relaxation of Assumption 1. For each negative example we draw $M$ step–size vectors $\alpha_i \in \mathcal{A}_i$ inside a pre-defined feasible set $\mathcal{F}$ (Assumption 2) and form augmented samples. If *none* of those augmented samples already scores above the target probability $p_{ex}$, the network is penalized with an additional explanation loss:

$$\mathcal{L}_{EXPL}(\theta_\phi, \theta_Y) := \frac{1}{N} \sum_{i=1}^{N} \left[ \mathcal{G}_i \cdot \text{BCE}(\underset{\alpha_i \in \mathcal{A}_i}{\text{avg}} \left[ Y_{\theta_Y}(\Phi_{\theta_\phi}(\underbrace{\mathbf{X}_i + \alpha_{\mathbf{X}_i}\delta_{\mathbf{X}_i}, \mathbf{O}_i + \alpha_{\mathbf{O}_i}\delta_{\mathbf{O}_i})}_{\text{CLEXNET's counterfactual edits 3}})) \right], p_{ex}) \right]. \tag{8}$$

This loss ensures that when changing the inputs following the given refusal reason, the model will predict an acceptance. The full loss combines representation balancing (Equation 6) and explanations:

$$\mathcal{L}_{CLEX}(\theta_\phi, \theta_Y, \theta_p) := \mathcal{L}_{BCE}(\theta_\phi, \theta_Y) - \lambda \cdot \mathcal{L}_{CE}(\theta_\phi, \theta_p) + \underbrace{\rho \cdot \mathcal{L}_{EXPL}(\theta_\phi, \theta_Y, \theta_p)}_{\text{CLEXNET's explanatory guidance 3}}, \tag{9}$$

where the multiplier $\rho \in [0, 1]$ serves to control between balancing the representation $\phi$, acceptance prediction and respecting the refusal reasons. CLEXNET's optimal parameters are then given by:

$$\theta_p^* := \underset{\theta_p}{\arg\max} \, \mathcal{L}_{CLEX}(\theta_\phi^*, \theta_Y^*, \theta_p) \quad \text{and} \quad (\theta_\phi^*, \theta_Y^*) := \underset{\theta_\phi, \theta_Y}{\arg\min} \, \mathcal{L}_{CLEX}(\theta_\phi, \theta_Y, \theta_p^*). \tag{10}$$

Finally, these parameters are learned by optimizing $\mathcal{L}_{CLEX}(\theta_\phi, \theta_Y, \theta_p)$ using stochastic gradient descent-based optimization and by placing a gradient reversal layer before the clustering head $c_{\theta_p}$. An instance-level training loop is shown in Algorithm 1, which deviates from a standard training loop in the case of a negative instance with a provided refusal reason.

Intuitively, the adversarial term removes information that stems from $\pi_{obs}$, while the explanation term injects causal directionality unavailable from labels alone. The two components thereby complement each other: without balancing, the model would learn confounded shortcuts; without explanations, it would be free to satisfy the loss in any direction, including biologically implausible ones.

**Implementation notes.**   In Equations 7 and 8, $\mathcal{A}_i$ represents the domain of feasible edit magnitudes for a specific instance. While $\mathcal{A}_i$ depends on $\mathbf{X}_i$ and $\mathbf{O}_i$, a more general feasible region $\mathcal{F} \subset \mathcal{X} \cup \mathcal{O}$ can be defined beforehand which can then be used to find $\mathcal{A}_i$ for every instance. Note that $\mathcal{F}$ need not coincide with the entire domain that would be explored in a gold-standard randomized controlled trial (RCT) [26]: practitioners are free to tighten the feasible region $\mathcal{F}$ by excluding biologically or logistically implausible edits based on their domain knowledge. Instead, our implementation constructs a hyper-box whose bounds are the per-feature minima and maxima observed in $\mathcal{D}$. Consequently, $\mathcal{A}_i$ is then constructed feature-wise for each instance. Finally, a random sampling method is used to sample $M$ step-sized $\alpha_i$ vectors.

The confidence in the explanations is represented by $p_{ex}$: if there exists a sample along direction $\delta$ that achieves acceptance probability $p_{ex}$, then $\mathcal{L}_{EXPL}$ becomes zero for that instance, otherwise, all sampled instances trigger a higher $\mathcal{L}_{EXPL}$ component.

# 4   Related work

**ML for organ allocation.**   Current U.S. match-runs still hinge on simple urgency scores—most prominently MELD [39] and MELD-Na[34]—that ignore how an organ's quality, competing patients, and logistics jointly shape long-term benefit. Motivated by these limitations, a growing line of work designs machine-learning-driven allocation rules. Causal policies rank candidates by estimated individual treatment effect or counterfactual survival [43, 66, 65, 8, 6], aiming to disentangle medical need from policy-induced confounding. A second stream enriches the objective with operational constraints such as transport distance, fairness and cold-ischemia limits, turning allocation into a combinatorial optimization problem, with often no closed-form solution [4, 11, 9, 45, 40]. More recent hybrids tackle both aspects simultaneously, coupling causal value estimates with stochastic models of future organ and patient arrivals so that present-day decisions account for tomorrow's opportunities [8, 6, 7]. However, apart from one approach [40], no prior work has embedded an acceptance estimator within the allocation policy itself—let alone one grounded in a causal framework.

**Learning with domain knowledge.**   While not being mutually exclusive, learning from the observed refusal reasons differs from injecting expert knowledge as inductive bias into a model: the former are empirical, data-driven labels or features obtained from real clinical decisions, whereas the latter entails *imposing* pre-defined domain rules or structures (e.g. hard-coding a donor age cutoff or penalty) into the learning process instead of letting the model discover such patterns from data [17, 60]. While incorporating domain knowledge can improve sample-efficiency, generalization and interpretability [31, 63], it introduces notable challenges. Eliciting, formalizing and maintaining expert rules is costly and time-consuming—the long-recognized "knowledge-acquisition bottleneck" of expert systems [53, 32, 14]. Moreover, relying on domain knowledge risks embedding incorrect or outdated assumptions: if an expert's belief is flawed, encoding it can mislead the model and remain hidden until it causes failures [3, 58, 16]. Consequently, leveraging the recorded refusal reasons—grounded in actual outcomes—offers a complementary and often safer causal signal, whereas expert-derived inductive biases must be applied judiciously and subjected to continuous post-deployment validation.

# 5   Experiments

Experiments on the choice of the feasible region $\mathcal{F}$, additional results for Experiment 5.1, and findings regarding empirical support for Assumption 1 can be found under Appendix A. More detailed information about experimental setups, synthetic functions, generation of refusal reasons and used features can be found in Appendix B. Information regarding hyperparameters and an acknowledgments section can be found in Appendix C and D respectively.

All code, synthetic generators and an implementation of CLEXNET are made public to facilitate independent assessment: `https://github.com/AlessandroMarchese/ClexNet`.

## 5.1   Does CLEXNET generalize better?

**Experimental setup.**   To evaluate the performance of different acceptance estimators we create two synthetic datasets that consist of patient-organ offer pairs: $\mathcal{D}_{obs}$ and $\mathcal{D}_{\mathcal{F}}$, where $\mathcal{D}_{obs}$ represents the

observational dataset which is affected by selection bias and $\mathcal{D}_{\mathcal{F}}$ represents an unbiased dataset with random patient-organ pairs, which is used to test the models in an unbiased way. $\mathcal{D}_{obs}$ is then split further into $\mathcal{D}_{train}$, which is used to train the models, and $\mathcal{D}_{test}$, which is used to test the models on observational data. We evaluate our model on both synthetic and semi-synthetic experiments.

For the synthetic evaluation, we construct a synthetic binary outcome function following a similar methodology to previous work in individual treatment effect estimation [8, 2, 27, 30, 52]: $f_Y(\mathbf{X}, \mathbf{O}) = \frac{1}{1+v\exp(h(\mathbf{X},\mathbf{O}))+\mathcal{N}_Y}$, where $h(\mathbf{X}, \mathbf{O})$ is a random non-linear function and $v$ is a scalar which is set such that the average acceptance probability corresponds with a chosen acceptance ratio. $Y$ is then sampled from a Bernoulli distribution with probability $f_Y(\mathbf{X}, \mathbf{O})$. Finally, refusal reasons are generated for all negative samples by finding the direction of the necessary edit such that $f_Y(\mathbf{X} + \alpha_{\mathbf{X}}\delta_{\mathbf{X}}, \mathbf{O} + \alpha_{\mathbf{O}}\delta_{\mathbf{O}})$ reaches at least some predefined probability $p_{ex}$.

For the semi-synthetic evaluation, we use UNOS-PTR [15] liver offers recorded between 2021 and 2024. This data consists of approximately 1.1M offers made between 24k unique organs and 46k unique patients. We considered refusal reasons regarding donor age and cold ischemic time. The used patient and organ features are shown in Appendix B.

**Benchmarks.** The performance of CLEXNET is compared against other traditional ML estimators from previous work [33, 10] and used in official simulators [54, 55, 56, 18, 19]. We also compare CLEXNET against other neural estimators: a single-task adaptation of PATIENTNET [40] and an ORGANITE model [8], adapted for binary classification.

Table 1: **Performance metrics on acceptance.** Models are trained and tested on both synthetic and semi-synthetic datasets. BCE, AUC, AUPRC and Brier score are evaluated on the test sets $\mathcal{D}_{\mathcal{F}}$. Standard deviations are instance-based and are shown in brackets. Models are ranked from least to most complex. Performances on the observational test sets can be found in Appendix A.

| Model | Synthetic data | | | | Semi-synthetic UNOS-PTR data | | | |
|---|---|---|---|---|---|---|---|---|
| | **BCE** | **AUC** | **AUPRC** | **Brier** | **BCE** | **AUC** | **AUPRC** | **Brier** |
| Logistic Regression [33] | .840 (.685) | .597 | .816 | .298 (.255) | 1.265 (2.536) | .540 | .279 | .237 (.390) |
| Random Forest [33, 10] | .543 (.313) | .737 | .889 | .182 (.139) | 2.195 (6.829) | .536 | .268 | .236 (.400) |
| PATIENTNET [40] | .784 (1.259) | .799 | .920 | .230 (.355) | .828 (1.348) | .593 | .308 | .233 (.397) |
| ORGANITE [8] | .442 (.696) | .840 | .932 | .140 (.234) | .855 (1.402) | .598 | .308 | .235 (.402) |
| CLEXNET ($\lambda = 0$) | .427 (.799) | .845 | .938 | .132 (.263) | .539 (.487) | .655 | .355 | .181 (.190) |
| CLEXNET | **.377** (.690) | **.872** | **.948** | **.117** (.221) | **.514** (.514) | **.704** | **.408** | **.170** (.206) |

+ Balancing
+ Explanations
+ Explanations

**Results.** The results are shown in Table 1. The single-task PATIENTNET, which ignores treatment-specific balancing, struggles to generalize beyond the biased training distribution, whereas ORGANITE's domain-invariant design recovers a sizeable performance boost. Building on the same balancing idea while further leveraging direction-only refusal information, CLEXNET consistently tops the table: it achieves the lowest error, the best ranking ability, and the sharpest probability calibration, demonstrating that coupling causal balancing with explanation-guided augmentation yields the most transferable acceptance model.

## 5.2 Does confounding affect performance?

**Experimental setup.** We test how robust CLEXNET is under different levels of confounding compared to other models. The same synthetic setup is used as in Experiment 5.1. However, the level of linear bias $\psi$ is gradually increased meaning that organs in $\mathcal{D}_{obs}$ become more dependent on the corresponding observed patients.

**Results.** Figure 4 plots performance metrics against an increasing linear confounder level $\psi$. PatientNet is the most vulnerable and reveals a substantial loss of accuracy and calibration with increasing bias. ORGANITE & CLEXNET are comparatively stable thanks to their balancing mechanism. Their metrics do drift upward/downward as bias strengthens, but the changes are modest and far smaller than those seen for PATIENTNET, showing that balancing—and for CLEXNET, the additional direction-only supervision—effectively dampens the impact of confounding.

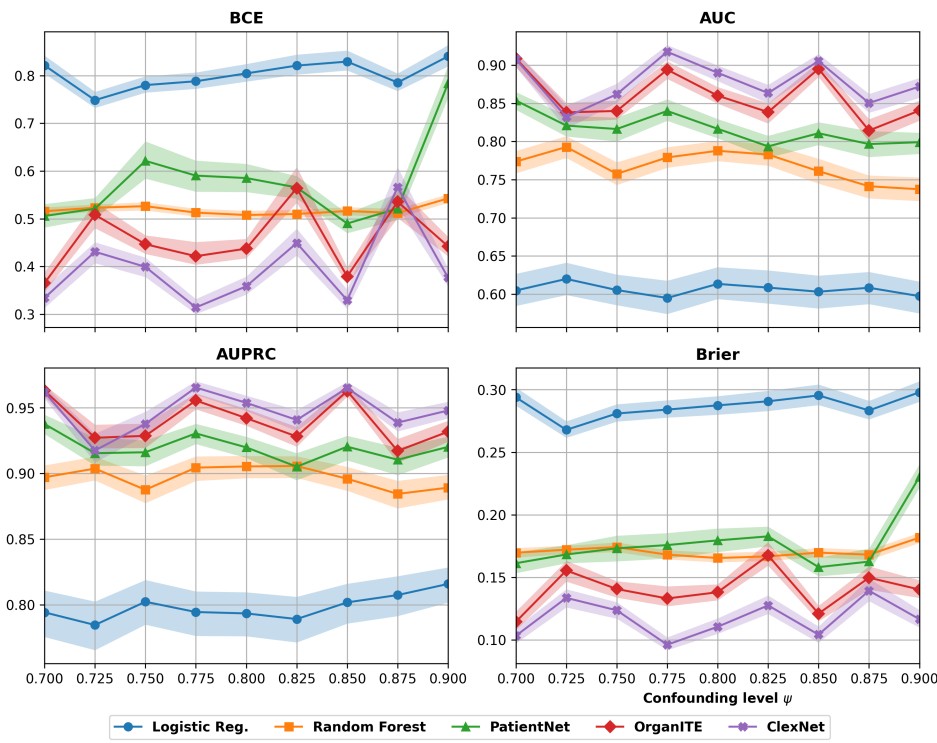

Figure 4: **Robustness of CLEXNET to increasing confounding.** Linear confounding $\psi$ is gradually introduced to add bias into the patient-organ pairs that are present in $\mathcal{D}_{obs}$. The models are tested on $\mathcal{D}_{\mathcal{F}}$. The bands represent 95% bootstrap percentile confidence intervals for each model. More information about the confounding mechanism can be found in Appendix B.

## 5.3 Are all directional reasons equally informative?

**Experimental setup.** We vary the mechanism that selects positive counterfactual samples paired with negative observations to generate directional refusal reasons $\delta_i$. Three mechanisms are compared: (i) random sampling from an unbiased set of feasible patient–organ pairs; (ii) inverse probability weighting (IPW), which reweights samples using a kernel density estimate $\hat{p}_{obs}(\mathbf{X}, \mathbf{O})$; and (iii) a boundary intersection method that selects counterfactuals crossing the decision boundary in low-density regions. A separate CLEXNET model is trained on the explanations generated by each mechanism.

**Results.** Even with explanations, the quality of the refusal-reason generation matters (Table 2). Random uniform sampling often samples from within the relatively high-density observational region. Although IPW favors positive counterfactuals in underrepresented regions, it does not guarantee that the decision boundary will not be crossed in the high-density observational region. Instead, the boundary intersection sampler tries to generate counterfactuals whose directions contain information that is not already contained in the observed distribution. Hence, effort spent on gathering plausible counterfactual directions is repaid in out-of-distribution performance.

Table 2: **Impact of different reason generation mechanisms on CLEXNET.** All metrics are evaluated on the unbiased set $\mathcal{D}_{\mathcal{F}}$. Instance-based standard deviations are shown in brackets.

| Reason mechanism | BCE | AUC | AUPRC | Brier |
|---|---|---|---|---|
| Uniform Random | .401 (.752) | .855 | .940 | .123 (.233) |
| IPW | .405 (.649) | .841 | .930 | .126 (.222) |
| Boundary Intersection | **.368 (.750)** | **.889** | **.958** | **.114 (.233)** |

# 6 Discussion

## 6.1 Limitations

While CLEXNET closes several gaps in current organ–offer modeling, important caveats remain.

**Dependence on the refusal–vector mapping $\mathcal{M}$.**   Our framework assumes a predefined mapping from each categorical refusal code to a signed direction vector $\delta = \mathcal{M}(R)$. In practice, some refusal reasons are coarse and represent latent features, requiring edits to a predefined set of multiple features that define the latent feature (e.g. "poor donor quality"). Any systematic misspecification of $\mathcal{M}$ will bias the explanation loss in Equation 8, potentially driving the decision boundary in an implausible direction.

**Feasibility region $\mathcal{F}$ and edit set $\mathcal{A}_i$.**   We bound counterfactual edits by a *hyper-box* whose edges are the feature-wise minima and maxima observed in $\mathcal{D}_{obs}$. This choice is intentionally conservative but imperfect: (i) it still permits biologically infeasible edits that are purely numerical outliers and (ii) it omits latent constraints such as blood-type compatibility or size-matching rules that are not explicit features. A misspecified $\mathcal{F}$ can either suppress the explanation loss (if too large) or force the model into an unreachable region (if too small; see Experiment A.1). Embedding clinical constraints (or a learned generative prior) into the augmentation sampler remains a challenge.

**(Semi-)Synthetic evaluation.**   All quantitative experiments are performed on controlled synthetic or semi-synthetic data sets whose generative mechanisms match the modeling assumptions (directional explanations refer to unobserved counterfactuals, $\mathcal{F}$ can be reasonably approximated by a hyper-box around $\mathcal{D}_{obs}$, etc.). Real data exhibit additional noise sources: missingness, time-varying policies, and outdated measurements. Deploying CLEXNET in an actual simulator therefore requires (i) auditing its calibration and fairness on historical wait-list snapshots and (ii) stress-testing under counterfactual policy shifts [7].

**Distributional changes are likely.**   The data-generating process that underpins CLEXNET is not static. Organ supply trends, recorded patient data, refusal codes, clinical practices, and policy rules all evolve over time, leading to *exogenous* distribution shifts. Moreover, once a model-based allocation policy is implemented, clinicians may change their acceptance behavior in response to the new incentives (*performative prediction*)—an *endogenous* shift created by the model itself [46]. Both phenomena can erode calibration, bias subgroup performance, and invalidate causal assumptions if left unchecked. Routine drift detection, scheduled re-evaluation of the refusal–vector mapping $\mathcal{M}$, and prospective shadow evaluation on fresh wait-list data are therefore mandatory safeguards before, during, and after deployment.

While the first two limitations are CLEXNET-*specific* and aim to weave richer domain knowledge directly into the architecture; the latter two address long-standing open problems for *all* causal models used in organ allocation [66, 65, 8, 6, 7, 40]. Addressing these limitations constitutes an important research agenda before CLEXNET can reliably inform real-world organ allocation policy.

## 6.2 Future work

**Embedding richer refusal information.**   Language-model embeddings could translate each coded or free-text refusal into a vector that lives in the same space as patient and donor features. An alignment layer could then connect that vector to the model's gradients, producing soft weights over which attributes should change—extending beyond simple directions and accommodating distance or monotonic hints. Treating the text-to-edit map as learnable replaces the predefined lookup table, opens the doors to free-text refusal reasons and lets uncertainty in explanations flow through training, effectively offering a continuous, instance-based relaxation of Assumption 1.

**Refining the feasible region.**   A practical edit domain could be built by shrinking the hyper-box to the high-density core of the observed patient-donor distribution and discarding segments that violate hard clinical rules such as blood type or size compatibility. This density-trimmed, constraint-aware region screens out implausible counterfactuals while keeping unbiased coverage of realistic cases, sharpening the explanation loss and improving data efficiency.

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

# A Additional results and experiments

## A.1 What role does $\mathcal{F}$ play?

As explained in Section 3, the role of $\mathcal{F}$ is crucial for the construction of $\mathcal{A}_i$. Without $\mathcal{F}$, the model would not know where to stop looking for plausible counterfactuals. CLEXNET constructs this feasible region $\mathcal{F}$ by storing the maximum and minimum values according to the features on the observational dataset $\mathcal{D}_{obs}$ to create $\mathcal{F}$, resulting in a hyper-box. In this experiment, we evaluate how well that works and look at other greedy and conservative approaches.

**Experimental setup.** The construction of the feasible region $\mathcal{F}$ is adapted to allow for its expansion or contraction. Let $f_{\min}$ and $f_{\max}$ denote the minimum and maximum observed values of a specific feature in $\mathcal{D}_{obs}$, and let $f_\mu$ be its observed mean. The bounds of the hyper-box are redefined feature-wise as:

$$f_{\min}^{(\sigma)} := f_\mu + \sigma(f_{\min} - f_\mu) \quad \text{and} \quad f_{\max}^{(\sigma)} := f_\mu + \sigma(f_{\max} - f_\mu). \tag{11}$$

where $\sigma$ is a scaling factor that expands ($\sigma > 1$) or contracts ($0 < \sigma < 1$) the region around $f_\mu$. Thus, by varying $\sigma$, CLEXNET is evaluated under different feasible region sizes $\mathcal{F}^{(\sigma)}$.

**Results.** The results are shown in Figures 6 and 5. CLEXNET's performance worsens significantly as $\mathcal{F}$ is contracted by decreasing $\sigma$. When $\mathcal{F}$ is slightly expanded, the performance mostly remains stable. Thus, underestimating $\mathcal{F}$ is worse than overestimating it.

Intuitively, when $\mathcal{F}$ is wrongly contracted, the decision boundary in unobserved regions gets wrongfully restricted, causing a significant drop in performance. In Figure 1, for example, the decision boundary would be squeezed between the observational region and $\mathcal{F}$. Instead, choosing a $\mathcal{F}$ that is too large simply diminishes the explanatory signal in Equation 8 rather than wrongly constraining the model. However, if $\mathcal{F}$ is set too large (such as in Figure 5), the signal from the explanations becomes less useful.

Figure 5: **The effects of changing $\mathcal{F}$ on CLEXNET's performance.** Performance metrics on the unbiased test set $\mathcal{D}_{\mathcal{F}}$ are shown for CLEXNET with different constructions of $\mathcal{F}$. These constructions are achieved by changing the scaling factor $\sigma$ that expands or contracts the original region $\mathcal{F}$ (at $\sigma = 1$, marked by a vertical blue dashed line). The bands represent 95% bootstrap percentile confidence intervals.

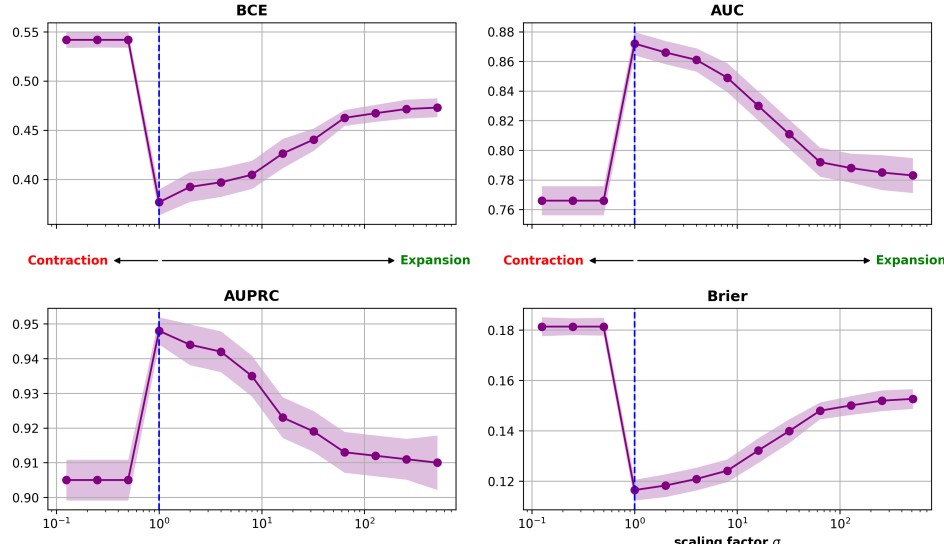

Figure 6: **The effects of changing $\mathcal{F}$ by magnitudes on CLEXNET's performance.** Performance metrics on the unbiased test set $\mathcal{D}_{\mathcal{F}}$ are shown for CLEXNET with different constructions of $\mathcal{F}$. These constructions are achieved by changing the scaling factor $\sigma$ that expands or contracts the original region $\mathcal{F}$ (at $\sigma = 1$, marked by a vertical blue dashed line). The bands represent 95% bootstrap percentile confidence intervals.

## A.2 Observational test set performance

Results on the observational test set $\mathcal{D}_{test}$ from Experiment 5.1 are reported in Table 3. The results show that PATIENTNET and ORGANITE perform better on the synthetic observational test set as they are less constrained than CLEXNET. On real observational data, Random Forest outperforms the other models. However, as shown in Table 1, CLEXNET is able to generalize significantly better on the unbiased test sets $\mathcal{D}_{\mathcal{F}}$.

Table 3: **Performance metrics on acceptance (observational test set).** Models are trained and tested on both synthetic and semi-synthetic datasets. BCE, AUC, AUPRC and Brier score are evaluated on the observational test sets $\mathcal{D}_{test}$. Standard deviations are shown in brackets.

| Model | Synthetic data | | | | Semi-synthetic UNOS-PTR data | | | |
|---|---|---|---|---|---|---|---|---|
| | BCE | AUC | AUPRC | Brier | BCE | AUC | AUPRC | Brier |
| Logistic Regression [33] | .707 (.388) | .590 | .558 | .254 (.172) | **.101** (.547) | .826 | .102 | .024 (.131) |
| Random Forest [33, 10] | .359 (.346) | .941 | .947 | .107 (.145) | .107 (.974) | **.867** | **.164** | **.022** (.125) |
| PATIENTNET [40] | **.229** (.694) | .969 | .970 | **.063** (.204) | .118 (.479) | .712 | .052 | .024 (.140) |
| ORGANITE [8] | .243 (.289) | **.979** | **.977** | .068 (.114) | .116 (.495) | .640 | .052 | .024 (.142) |
| CLEXNET ($\lambda = 0$) | .236 (.576) | .962 | .961 | .066 (.193) | .158 (.348) | .626 | .042 | .032 (.123) |
| CLEXNET | .275 (.367) | .970 | .958 | .076 (.129) | .159 (.337) | .598 | .036 | .030 (.123) |

## A.3 Empirical support for Assumption 1

Although CLEXNET uses a relaxed version of Assumption 1, it is possible to test empirical support for this assumption in the real data.

**Experimental setup.** To test support for Assumption 1, we try to match each refusal, with corresponding refusal reason, to a similar acceptance that i) satisfies that reason and ii) is within a certain range of the refusal.

Table 4: **Support for Assumption 1 versus allowed matching range.** Percentages shown are the cumulative support for donor-age–related reasons, cold–ischemic-time–related reasons, and overall, as the maximum Euclidean-distance threshold increases.

| Allowed matching range (Euclidean distance) | Support for donor age related reasons | Support for cold ischemic time related reasons | Overall Support |
|---|---|---|---|
| $\leq 1$ | 0.1% | 0.1% | 0.1% |
| $\leq 2$ | 7.0% | 4.7% | 6.2% |
| $\leq 3$ | 71.7% | 68.8% | 70.7% |
| $\leq 4$ | 98.7% | 97.8% | 98.4% |
| $\leq 5$ | 99.9% | 99.8% | 99.9% |
| $\leq 6$ | 100.0% | 100.0% | 100.0% |

**Results.** If we allow for matching refusals with acceptances within a Euclidean distance of 3 (considering that the feature space has 76 dimensions after OHE), we can find suitable positives that satisfy Assumption 1 for over 70% of the observed refusals with reasons related to donor age or cold ischemic time.

# B   Experimental setups

## B.1   Experimental setup for all synthetic studies

This section spells out every stochastic component, hyper-parameter and practical decision that enters the construction of the synthetic data sets used throughout all experiments. Re-implementing the pipeline line-by-line should therefore reproduce the raw data on which CLEXNET and the baselines were trained and evaluated.

**Notation recap.**   A single observation is a quadruplet $(\mathbf{X}, \mathbf{O}, Y, R)$ where

- $\mathbf{X} \in \mathbb{R}^{d_x}$ — patient covariates (demographic, clinical, logistical),
- $\mathbf{O} \in \mathbb{R}^{d_o}$ — organ–offer attributes (donor and procurement information),
- $Y \in \{0, 1\}$ — acceptance indicator, 1 represents accepted,
- $R \in \mathcal{R}$ — categorical refusal reason when $Y = 0$.

We fix $d_x = d_o = 5$ in all experiments.

### B.1.1   Generating patient covariates X

Patients are sampled i.i.d. from a standard multivariate normal:

$$\mathbf{X} \sim \mathcal{N}(\mathbf{0}, I_{d_x}). \tag{12}$$

This choice deliberately avoids introducing any implicit structure; all correlations subsequently arise from the confounding mechanism.

### B.1.2   Generating organ offers O with tunable confounding

To model the clinical intuition that organs are *not* allocated independently of the candidates to whom they are offered, we introduce a *linear confounding parameter* $\psi \in [0, 1)$ and draw

$$\mathbf{O} = \psi \mathbf{X} \mathbf{A}^\top + \sqrt{1 - \psi^2}\,\boldsymbol{\varepsilon} \qquad \boldsymbol{\varepsilon} \sim \mathcal{N}(\mathbf{0}, I_{d_o}), \quad \mathbf{A}_{pq} \sim \mathcal{N}\!\left(0, \frac{1}{\sqrt{d_x}}\right). \tag{13}$$

When $\psi = 0$ organs and patients are independent; as $\psi \to 1$ they become almost deterministically aligned via a random matrix $\mathbf{A}$ (Experiment 5.2 sweeps across $\psi$).

Moreover, an additional, non-linear bias is added by following the procedure:

1. draw an i.i.d. pool of $N_0$ candidate pairs $(\mathbf{X}, \mathbf{O})$ using Steps B.1.1–B.1.2 (with $\psi = 0$);

2. pass every pair through a random, non-linear function $g(\mathbf{X}, \mathbf{O})$ (randomly initialized, frozen thereafter) and retain the scalar score $s_i = g(\mathbf{X}_i, \mathbf{O}_i)$;

3. keep only the top 5% and bottom 5% of pairs by $s_i$.

The retained 10% constitute the observational data set:

$$\mathcal{D}_{obs} = \big((\mathbf{X}_i, \mathbf{O}_i) : s_i \text{ in extreme decile}\big), \qquad N := |\mathcal{D}_{obs}| \approx 0.1 N_0.$$

**Train/test split.** We allocate 70% of $\mathcal{D}_{obs}$ to $\mathcal{D}_{train}$, 15% for a validation set and 15% to $\mathcal{D}_{test}$ (stratified by $Y$).

### B.1.3 Ground-truth outcome mechanism $f_Y$

Acceptance probability is a random logistic transformation of a frequently used scoring function with an additional interaction term [8, 2, 27, 30, 52]:

$$h(\mathbf{X}, \mathbf{O}) = \mathbf{w}_1 \mathbf{X} + \mathbf{w}_2 \mathbf{O} + \mathbf{X}^\top \mathbf{W}_3 \mathbf{O}, \tag{14}$$

$$f_Y(\mathbf{X}, \mathbf{O}) = \frac{1}{1 + v \exp\left(h(\mathbf{X}, \mathbf{O})\right) + \mathcal{N}_Y} \tag{15}$$

Here, $\mathbf{w}_1$ and $\mathbf{w}_2$ represent random vectors, and $\mathbf{W}_3 \in \mathbb{R}^{d_x \times d_o}$ is a matrix with random entries. The scalar $v$ is chosen such that $\mathbb{E}_{\mathcal{D}_{obs}}[f_Y] \approx 0.50$, making acceptance a balanced label. Finally, outcomes are sampled such that $Y \sim \text{Bernoulli}\big(f_Y(\mathbf{X}, \mathbf{O})\big)$.

### B.1.4 Generating direction-only refusal reasons $R$

For every negative instance $Y = 0$ we synthetically attach a *direction* $\delta$ rather than an absolute counterfactual. The procedure is as follows:

1. Draw an auxiliary set $\mathcal{D}_{\mathcal{F}}^{\delta}$ of $(\mathbf{X}', \mathbf{O}')$ pairs without the bias.
2. Keep only those candidates with $f_Y(\mathbf{X}', \mathbf{O}') \geq p_{ex}$ (default $p_{ex} = 0.5$).
3. Uniformly sample one such "positive" and compute $\delta := (\mathbf{X}' - \mathbf{X}, \mathbf{O}' - \mathbf{O})$, then map $\delta$ onto $\Delta$ based on the sign of each element.
4. (Optional) Map $\delta$ to a categorical refusal label $R$ via a lookup table $\mathcal{M}^{-1}$ (many-to-one). The last step is bypassed in the experiments and $\delta$ is used directly. The model never sees magnitude information since delta mapped onto $\Delta$.

**Feasible domain $\mathcal{F}$.** We store the per-feature minima and maxima over *all* draws (before selection). Sampling magnitudes $\alpha \in \mathcal{A}_i$ then clamps each edited feature to this hyper-box to respect Assumption 2.

## B.2 Experimental setup for semi-synthetic studies

Liver offers from the UNOS-PTR [15] dataset, recorded between 2021 and 2024, are used for the semi-synthetic evaluation of CLEXNET. Following previous work [33], only offers related to organs that eventually got placed are considered, resulting in approximately 1.1M offers. The considered features are shown in Table 5. For the construction of $\mathcal{D}_{\mathcal{F}}$, a separate CLEXNET model was trained on the observational data (including the refusal reasons) and used as $f_Y(\mathbf{X}, \mathbf{O})$. Next, real patients are paired with real organs to generate $\mathcal{D}_{\mathcal{F}}$. This way, the real covariate structures of patients and organs are preserved.

## B.3 Compute resources

Experiments ran on a 13th Gen Intel(R) Core(TM) i9-13900HX processor with 32GB RAM. End-to-end wall-clock time: training CLEXNET $\approx 60$s per run, including the in-the-loop explanation-guided augmentation (Algorithm 1).

Table 5: Considered features from the UNOS-PTR dataset [15] to represent patients and organs.

| Patient Features | |
|---|---|
| GENDER | recipient gender |
| DAYSWAIT_CHRON | days on liver waiting list |
| ETHCAT | recipient ethnicity category |
| INIT_AGE | age in years at time of listing |
| INIT_ALBUMIN | initial waiting list albumin |
| INIT_ASCITES | initial waiting list ascites |
| INIT_BMI_CALC | calculated candidate bmi at listing |
| INIT_BILIRUBIN | initial waiting list bilirubin |
| INIT_INR | initial waiting list inr |
| INIT_SERUM_CREAT | initial waiting list serum creatinine |
| INIT_SERUM_SODIUM | initial waiting list serum sodium |
| HGT_CM_CALC | calculated recipient height (cm) |
| WGT_KG_CALC | calculated recipient weight (kg) |
| DIAG | recipient primary diagnosis |
| **Organ Features** | |
| AGE_DON | donor age (yrs) |
| ALCOHOL_HEAVY_DON | ddr heavy alcohol use (heavy= 2+ drinks/day) (y/n/u) |
| BMI_DON_CALC | donor bmi - pre/at donation calculated |
| COD_CAD_DON | deceased donor-cause of death |
| COLD_ISCH | total cold ischemic time (hours) |
| ETHCAT_DON | donor ethnicity category |
| HGT_CM_DON_CALC | calculated donor height (cm) |
| HIST_CANCER_DON | deceased donor-history of cancer (y/n) |
| HIST_CIG_DON | deceased donor-history of cigarettes in past (more than 20 pack yrs) |
| GENDER_DON | donor gender |
| NON_HRT_DON | deceased donor-non-heart beating donor |

# C CLEXNET's hyperparameters

The hyperparameters that were used for CLEXNET in the experiments can be found in Figure 7. The same hyperparameters are used for $\Phi_{\theta_\phi}(\mathbf{X}, \mathbf{O})$, $Y_{\theta_Y}(\phi)$ and $c_{\theta_p}(\phi)$ across all neural network based models. Following previous work [8], the organ clustering function $c(\cdot)$ and thus the organ clusters $c_i$ are determined by a k-means clustering algorithm.

Two studies are shown in which novel hyperparameters are varied:

1. In Table 6 variations of CLEXNET with different values for $\lambda$ and $\rho$ are shown. These parameters control the trade-offs between predictions on the observational data, representation balancing and respecting explanations. Both parameters significantly impact the performance on $\mathcal{D}_F$.

2. On Figure 7 variations of CLEXNET with different values of $M$ are shown. This parameter controls how many augmented data points need to be sampled in the training loop (Algorithm 1) for each instance with a refusal reason. The results show that the benefit of increasing $M$ flattens after a certain magnitude is reached ($M = 100$ in this case).

Table 6: **CLEXNET's performance with different loss weights.** CLEXNET variants are shown with different weights for loss components $\lambda$ (for representation balancing) and $\rho$ (for explanatory supervision). The blue cells correspond to the results from Experiment 5.1. Standard deviations are calculated using bootstraps and are shown in brackets.

| BCE on $\mathcal{D}_{test}$ | | | | |
|---|---|---|---|---|
| $\lambda \setminus \rho$ | 0.05 | 0.10 | 0.15 | 0.20 |
| 0.05 | **.225 (.027)** | .231 (.026) | .236 (.028) | .243 (.034) |
| 0.10 | .243 (.023) | .250 (.025) | .255 (.029) | .248 (.024) |
| 0.15 | .276 (.026) | .281 (.027) | .275 (.031) | .281 (.032) |
| 0.20 | .284 (.032) | .293 (.030) | .296 (.027) | .297 (.025) |

| BCE on $\mathcal{D}_{\mathcal{F}}$ | | | | |
|---|---|---|---|---|
| $\lambda \setminus \rho$ | 0.05 | 0.10 | 0.15 | 0.20 |
| 0.05 | .431 (.008) | .400 (.009) | .419 (.009) | .446 (.009) |
| 0.10 | .405 (.007) | .411 (.007) | .396 (.007) | .396 (.009) |
| 0.15 | .405 (.007) | .405 (.006) | **.377 (.007)** | .384 (.007) |
| 0.20 | .390 (.007) | .383 (.007) | .392 (.008) | .395 (.006) |

| AUROC on $\mathcal{D}_{test}$ | | | | |
|---|---|---|---|---|
| $\lambda \setminus \rho$ | 0.05 | 0.10 | 0.15 | 0.20 |
| 0.05 | **.978 (.010)** | .975 (.010) | .973 (.011) | .969 (.013) |
| 0.10 | .977 (.010) | .976 (.010) | .975 (.012) | .976 (.011) |
| 0.15 | .974 (.011) | .972 (.012) | .970 (.014) | .966 (.015) |
| 0.20 | .965 (.016) | .963 (.015) | .967 (.014) | .973 (.012) |

| AUROC on $\mathcal{D}_{\mathcal{F}}$ | | | | |
|---|---|---|---|---|
| $\lambda \setminus \rho$ | 0.05 | 0.10 | 0.15 | 0.20 |
| 0.05 | .841 (.004) | .880 (.004) | .869 (.004) | .861 (.004) |
| 0.10 | .854 (.004) | .849 (.004) | .865 (.004) | **.873 (.004)** |
| 0.15 | .845 (.004) | .841 (.005) | .872 (.004) | **.873 (.004)** |
| 0.20 | .865 (.004) | .869 (.004) | .860 (.004) | .853 (.004) |

| AUPRC on $\mathcal{D}_{test}$ | | | | |
|---|---|---|---|---|
| $\lambda \setminus \rho$ | 0.05 | 0.10 | 0.15 | 0.20 |
| 0.05 | **.976 (.012)** | .972 (.011) | .970 (.013) | .964 (.017) |
| 0.10 | .975 (.012) | .973 (.012) | .972 (.015) | .972 (.013) |
| 0.15 | .968 (.016) | .967 (.017) | .958 (.024) | .957 (.022) |
| 0.20 | .949 (.030) | .947 (.027) | .953 (.026) | .965 (.021) |

| AUPRC on $\mathcal{D}_{\mathcal{F}}$ | | | | |
|---|---|---|---|---|
| $\lambda \setminus \rho$ | 0.05 | 0.10 | 0.15 | 0.20 |
| 0.05 | .932 (.003) | **.953 (.002)** | .947 (.002) | .944 (.002) |
| 0.10 | .940 (.003) | .937 (.003) | .945 (.002) | .950 (.002) |
| 0.15 | .933 (.003) | .931 (.003) | .948 (.002) | .948 (.002) |
| 0.20 | .943 (.003) | .945 (.003) | .941 (.003) | .937 (.003) |

| Brier on $\mathcal{D}_{test}$ | | | | |
|---|---|---|---|---|
| $\lambda \setminus \rho$ | 0.05 | 0.10 | 0.15 | 0.20 |
| 0.05 | **.066 (.011)** | .069 (.010) | .069 (.011) | .071 (.012) |
| 0.10 | .068 (.009) | .071 (.010) | .073 (.011) | .070 (.009) |
| 0.15 | .078 (.010) | .079 (.010) | .076 (.011) | .078 (.011) |
| 0.20 | .080 (.011) | .083 (.011) | .083 (.010) | .083 (.010) |

| Brier on $\mathcal{D}_{\mathcal{F}}$ | | | | |
|---|---|---|---|---|
| $\lambda \setminus \rho$ | 0.05 | 0.10 | 0.15 | 0.20 |
| 0.05 | .131 (.003) | .119 (.002) | .123 (.003) | .128 (.003) |
| 0.10 | .125 (.002) | .127 (.002) | .121 (.002) | .120 (.002) |
| 0.15 | .126 (.002) | .127 (.002) | **.116 (.002)** | .118 (.002) |
| 0.20 | .120 (.002) | .118 (.002) | .121 (.002) | .123 (.002) |

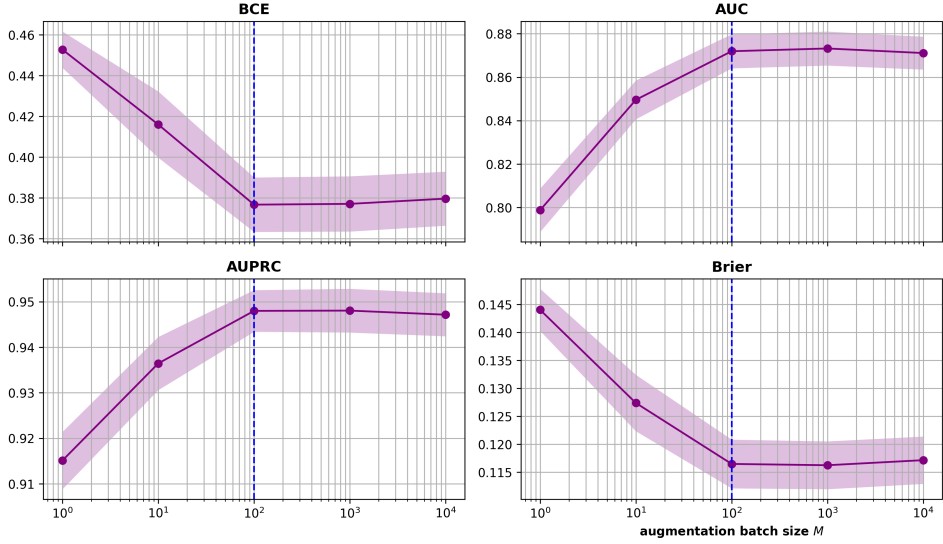

Figure 7: **CLEXNET's performance for different augmentation batch sizes.** Performance metrics on the unbiased test set $\mathcal{D}_{\mathcal{F}}$ are shown for CLEXNET with different values for the augmentation batch size $\mathcal{M}$. The default value used in other experiments is set at $\mathcal{M} = 100$ (marked by a vertical blue dashed line). The bands represent 95% bootstrap percentile confidence intervals.

Table 7: CLEXNET's Hyperparameters

| Component | Hyperparameters |
|---|---|
| **Shared Encoder** $\Phi_{\theta_\phi}(\mathbf{X}, \mathbf{O})$ | Dense(32, L2), ReLU Activation Dense(32, L2), ReLU Activation |
| **Acceptance Head** $Y_{\theta_Y}(\phi)$ | Dense(32, L2), Sigmoid Activation |
| **Organ Cluster Head** $c_{\theta_p}(\phi)$ | Dense(32, L2), ReLU Activation |
| **Organ Cluster Amount** $k$ | 3 |
| **Organ Cluster Loss Weight** $\lambda$ | 0.15 |
| **Explanation Loss Weight** $\rho$ | 0.15 |
| **Augmentation Batch** $M$ | 100 |
| **Training Parameters** | Maximum Epochs: 1000 Patience: 30 Learning Rate: $1 \times 10^{-3}$ |

# D  Acknowledgments

The organ-patient pair data reported in this work has been supplied by the United Network for Organ Sharing as the contractor for the Organ Procurement and Transplantation Network. The interpretation and reporting of the data are the responsibility of the author(s) and in no way should be seen as an official policy of or interpretation by the OPTN or the U.S. Government.

