# OpenReview forum: "Causal Explanation-Guided Learning for Organ Allocation"
_NeurIPS.cc/2025/Conference — NeurIPS 2025 poster_

### Official Review · Reviewer_iTSa · 2025-07-01

**Clarity:** 3
**Significance:** 3
**Originality:** 2
**Rating:** 5
**Confidence:** 2

**Summary:**

This paper proposes a method called CLEXNET for learning a model of organ offer acceptance. CLEXNET has two key components: 1) A scheme for learning balanced representations of organ clusters that removes observational bias via adversarial training, and 2) A data augmentation method and associated explanation loss for learning from refusal codes associated with organ offer rejections. On experiments with synthetic data, CLEXNET is shown to perform better (according to BCE, AUC, AUPRC, Brier score) than previous methods, including the same model without the explanation guided loss.

**Questions:**

- Why do experiments only use synthetic datasets, rather than natural data, even for the observational dataset (line 237)?
- Does the approach allow for interaction between the features cited in a contrastive counterfactual explanation? For example, what happens if two reasons are cited, and the reasons interact with each other to affect what the counterfactual outcome would have been (compared to if only a single reason were responsible for the organ rejection)?
- Which parts of CLEXNET are novel compared to previous work on learning models of organ-offer acceptance?

**Ethical Concerns:**

["NO or VERY MINOR ethics concerns only"]

**Final Justification:**

My main concern, about experiments with only synthetic datasets, has been addressed. Authors also addressed the weakness about it not being clear which parts are novel. I have raised my score accordingly.

**Limitations:**

yes

**Quality:**

2

**Strengths And Weaknesses:**

Strengths:
- The work is well-motivated: The introduction lays out the key challenges that the work aims to address and how previous works have ailed to address these challenges.
- The paper is generally well-organized (minus where the specific novel aspects of CLEXNET could be clarified -- see below).
- The general idea of treating refusal codes as counterfactual explanations to augment data is simple yet clever.

Weaknesses:
- All experiments are on synthetic datasets, calling into question the robustness of the results. For example, it is not clear whether the rules used to map features in counterfactual explanations to directions would generalize to real-world datasets, where explanations and their relationships to datasets would be more complex.
- The proposed method, CLEXNET, has several key steps, and it is unclear which are claimed to be novel (for example, just the method of learning from refusal codes, or also the balanced training step?). This could be clarified in the paper.

Minor Notes:
Line 102: "The ranking... the sequence of potential transplant recipients *who* are considered" --> missing "who"

---

> ### Author Response · Authors · 2025-07-31
> **Reviewer iTSa part 1**
>
> Dear Reviewer iTSa,
>
> Thank you for your careful reading of our manuscript. We appreciate your recognition of the motivation behind the work, the clarity of the presentation, and the potential of treating refusal codes as counterfactual cues. Below we respond point‑by‑point to each of your comments and describe the revisions now reflected in the updated paper.
>
> **Evaluation limited to synthetic data.**
> > *Actions taken:* We added semi-synthetic experiments using real‑world 2021-2024 UNOS-PTR liver‑offer data (n ≈ 1.1 M offers, 24k donors, 46k candidates).
>
> We ran additional experiments using real patients $X$ and real organs $O$ with the same synthetic setup presented in Appendix A.3. This way, the real covariate structures of patients and organs are preserved.
>
> | Model          | $\text{BCE}_\mathcal{F}$ | $\text{AUC}_\mathcal{F}$ | $\text{AUPRC}_\mathcal{F}$ | $\text{Brier}_\mathcal{F}$ |
> |----------------|------------------------:|-------------------------:|---------------------------:|---------------------------:|
> | Logistic Reg.  | 0.885 (0.672) | 0.588 | 0.771 | 0.319 (0.255) |
> | Random Forest  | 0.582 (0.275) | 0.774 | 0.891 | 0.199 (0.127) |
> | PatientNet     | 0.565 (0.963) | 0.842 | 0.920 | 0.177 (0.313) |
> | OrganITE       | 0.491 (0.828) | 0.848 | 0.922 | 0.156 (0.283) |
> | ClexNet        | **0.421 (0.662)** | **0.851** | **0.926** | **0.132 (0.234)** |
>
> Additionally, we ran experiments where every instance (consisting of a patient $X$, organ $O$, outcome $Y$ and refusal reason $R$) is real. To evaluate our models we considered the real dataset as $\mathcal{D}_{obs}$ and constructed the feasible set $\mathcal{D_F}$ by randomly rearranging real organs and real patients. We considered and encoded real refusal reasons regarding the donor age and cold ischemia time.
>
> | Model          | $\text{BCE}_\mathcal{F}$ | $\text{AUC}_\mathcal{F}$ | $\text{AUPRC}_\mathcal{F}$ | $\text{Brier}_\mathcal{F}$ |
> |----------------|------------------------:|-------------------------:|---------------------------:|---------------------------:|
> | Logistic Reg.  | 1.265 (2.536) | 0.540 | 0.279 | 0.237 (0.390) |
> | Random Forest  | 2.195 (6.829) | 0.536 | 0.268 | 0.236 (0.400) |
> | PatientNet     | 0.828 (1.348) | 0.593 | 0.308 | 0.233 (0.397) |
> | OrganITE       | 0.855 (1.402) | 0.598 | 0.308 | 0.235 (0.402) |
> | ClexNet        | **0.514 (0.514)** | **0.704** | **0.408** | **0.170 (0.206)** |
>
> >  *Actions taken:* We added a table that contains all used real features with a short explanation of what they represent in the appendix.
>
> | Patient Features   | Organ Features      |
> |--------------------|---------------------|
> | GENDER             | AGE_DON             |
> | DAYSWAIT_CHRON     | ALCOHOL_HEAVY_DON   |
> | ETHCAT             | BMI_DON_CALC        |
> | INIT_AGE           | COD_CAD_DON         |
> | INIT_ALBUMIN       | COLD_ISCH           |
> | INIT_ASCITES       | ETHCAT_DON          |
> | INIT_BMI_CALC      | HGT_CM_DON_CALC     |
> | INIT_BILIRUBIN     | HIST_CANCER_DON     |
> | INIT_INR           | HIST_CIG_DON        |
> | INIT_SERUM_CREAT   | GENDER_DON          |
> | INIT_SERUM_SODIUM  | NON_HRT_DON         |
> | HGT_CM_CALC        | DON_TY              |
> | WGT_KG_CALC        |                     |
> | DIAG               |                     |
>
> **Interactions between the features cited in a contrastive counterfactual explanation.**
> Yes, our approach supports this. This can occur in two cases which are not mutually exclusive: i) there are multiple refusal reasons which involve different features (this occurs in a minority of the cases) and ii) the provided refusal reason directly involves multiple features. In both cases, the counterfactual sampling space $\mathcal{A}_i$ becomes a hyperspace, making the reasons less informative.
>
> **Which parts of CLEXNET are novel compared to previous work.**
> We like to point out that current ML-based allocation studies sidestep the complexity of refusals, assuming every offer is accepted  [8,6,7,63,64]. while more clinically-oriented studies use models of limited complexity and neglect causal mechanisms [55,54,53,17,18].  In the context of organ offer acceptance, everything we present in our work is novel.
>
> In the context of ML, the main novelty is indeed the causal learning from refusal reasons by sampling counterfactuals while the domain adversarial learning for balancing representations has already been done in previous work [13,8].

---

> ### Author Response · Authors · 2025-07-31
> **Reviewer iTSa part 2**
>
> In our work, everything which is highlighted in light-blue is novel. This includes:
> * Figure 3: the counterfactual sampling parts and the explanation loss.
> * Algorithm 1: embedding of reasons, directional augmentation (Eq. 1, 2), forward pass of counterfactual, max and average of counterfactuals, initialize explanation loss, guarded explanation loss (Eq. 7, 8).
> * Eq. 7-9 which are the core of our methodology and represent the explanatory loss.
>
> > *Actions taken:* We made sure to clarify in the footnote that everything highlighted in light blue is novel and part of our main contribution.
>
> Your feedback helped us strengthen the empirical evaluation, clarify our contributions. We hope these additions address your concerns and demonstrate the practical utility and novelty of our approach.
>
> Sincerely,
> Anonymous Authors
>
> ---
>
> [6] Berrevoets, J., Alaa, A., Qian, Z., Jordon, J., Gimson, A. E., & Van Der Schaar, M. (2021, July). Learning queueing policies for organ transplantation allocation using interpretable counterfactual survival analysis. In International Conference on Machine Learning (pp. 792-802). PMLR.
>
> [7] Berrevoets, J., Jarrett, D., Chan, A., & van der Schaar, M. (2023). Allsim: Simulating and benchmarking resource allocation policies in multi-user systems. Advances in Neural Information Processing Systems, 36, 851-866.
>
> [8] Berrevoets, J., Alaa, A., Qian, Z., Jordon, J., Gimson, A. E., & Van Der Schaar, M. (2021, July). Learning queueing policies for organ transplantation allocation using interpretable counterfactual survival analysis. In International Conference on Machine Learning (pp. 792-802). PMLR.
>
> [13] Bica, I., Alaa, A. M., Jordon, J., & Van Der Schaar, M. (2020). Estimating counterfactual treatment outcomes over time through adversarially balanced representations. arXiv preprint arXiv:2002.04083.
>
> [17] de Ferrante, H., Rosmalen, M. D. R. V., Smeulders, B., Spieksma, F. C., & Vogelaar, S. (2024). A discrete event simulator for policy evaluation in liver allocation in Eurotransplant. arXiv preprint arXiv:2410.10840.
>
> [18] de Ferrante, H. C., Goya, R. L., Smeulders, B. M., Spieksma, F. C., & Tieken, I. (2025). The ETKidney simulator: a discrete event simulator to assess the impact of alternative kidney allocation rules in Eurotransplant. arXiv preprint arXiv:2502.15001.
>
> [53] SRTR. KPSAM 2015 User Guide, 2015.
>
> [54] SRTR. TSAM 2015 User Guide, 2015.
>
> [55] SRTR. LSAM 2019 User Guide, 2019
>
> [63] Wood, N. L., Mogul, D. B., Perito, E. R., VanDerwerken, D., Mazariegos, G. V., Hsu, E. K., ... & Gentry, S. E. (2021). Liver simulated allocation model does not effectively predict organ offer decisions for pediatric liver transplant candidates. American Journal of Transplantation, 21(9), 3157-3162.
>
> [64] Xu, C., Alaa, A., Bica, I., Ershoff, B., Cannesson, M., & van der Schaar, M. (2021, March). Learning matching representations for individualized organ transplantation allocation. In International Conference on Artificial Intelligence and Statistics (pp. 2134-2142). PMLR.

---

> ### Comment · Reviewer_iTSa · 2025-08-08
>
> Thank you for your thorough response. My main concern, about experiments with only synthetic datasets, has been addressed. I have raised my score accordingly.

---

### Official Review · Reviewer_AejF · 2025-07-03

**Clarity:** 3
**Significance:** 3
**Originality:** 3
**Rating:** 5
**Confidence:** 3

**Summary:**

The paper proposes a new machine learning method for organ allocation. The core idea is to utilize the categorical refusal reasons, which were not exploited in previous methods, to learn an embedding of counterfactual edits to the input features. Experiments on synthetic data show that the proposed model combined with this new source of information improves the acceptance rate over previous methods.

**Questions:**

None

**Ethical Concerns:**

["NO or VERY MINOR ethics concerns only"]

**Final Justification:**

The rebuttal addresses concerns and I keep my original positive rating.

**Limitations:**

Yes

**Quality:**

3

**Strengths And Weaknesses:**

**Strengths:**
- The method exploits a new source of information previously ignored by other methods: a categorical refusal reason (e.g. “donor too old”, “size mismatch”), which encodes rich information that can potentially improve the acceptance prediction.
- The refusal reason describes what feature causes the rejection, but does not indicate how much we should change that feature. The authors formulate an explanation loss to make the model obey the assumption that there exists some magnitude of change along the counterfactual direction to flip the prediction.
The presentation is clear, with easy-to-understand motivation and modeling details, for a reader not familiar with this topic to dive in.

**Weaknesses:**
- The paper uses synthetic data for all the experiments. I am not familiar enough with the norms of this field to know if this is widely accepted. The natural concern is the modeling method of the synthetic data may not fully reflect the complexity of the real-world problem, leading to biased conclusions.

---

> ### Author Response · Authors · 2025-07-31
> **Reviewer AejF**
>
> Dear Reviewer AejF,
>
> Thank you for taking the time to read our submission and for highlighting both the promise of leveraging refusal‑reason information and the limitations of our current evaluation. Below we respond to your points in detail and describe the concrete steps we have taken in the revised manuscript.
>
> **Evaluation limited to synthetic data.**
> > *Actions taken:* We added semi-synthetic experiments using real‑world 2021-2024 UNOS-PTR liver‑offer data (n ≈ 1.1 M offers, 24k donors, 46k candidates).
>
> We ran additional experiments using real patients $X$ and real organs $O$ with the same synthetic setup presented in Appendix A.3. This way, the real covariate structures of patients and organs are preserved.
>
> | Model          | $\text{BCE}_\mathcal{F}$ | $\text{AUC}_\mathcal{F}$ | $\text{AUPRC}_\mathcal{F}$ | $\text{Brier}_\mathcal{F}$ |
> |----------------|------------------------:|-------------------------:|---------------------------:|---------------------------:|
> | Logistic Reg.  | 0.885 (0.672) | 0.588 | 0.771 | 0.319 (0.255) |
> | Random Forest  | 0.582 (0.275) | 0.774 | 0.891 | 0.199 (0.127) |
> | PatientNet     | 0.565 (0.963) | 0.842 | 0.920 | 0.177 (0.313) |
> | OrganITE       | 0.491 (0.828) | 0.848 | 0.922 | 0.156 (0.283) |
> | ClexNet        | **0.421 (0.662)** | **0.851** | **0.926** | **0.132 (0.234)** |
>
> Additionally, we ran experiments where every instance (consisting of a patient $X$, organ $O$, outcome $Y$ and refusal reason $R$) is real. To evaluate our models we considered the real dataset as $\mathcal{D}_{obs}$ and constructed the feasible set $\mathcal{D_F}$ by randomly rearranging real organs and real patients. We considered and encoded real refusal reasons regarding the donor age and cold ischemia time.
>
> | Model          | $\text{BCE}_\mathcal{F}$ | $\text{AUC}_\mathcal{F}$ | $\text{AUPRC}_\mathcal{F}$ | $\text{Brier}_\mathcal{F}$ |
> |----------------|------------------------:|-------------------------:|---------------------------:|---------------------------:|
> | Logistic Reg.  | 1.265 (2.536) | 0.540 | 0.279 | 0.237 (0.390) |
> | Random Forest  | 2.195 (6.829) | 0.536 | 0.268 | 0.236 (0.400) |
> | PatientNet     | 0.828 (1.348) | 0.593 | 0.308 | 0.233 (0.397) |
> | OrganITE       | 0.855 (1.402) | 0.598 | 0.308 | 0.235 (0.402) |
> | ClexNet        | **0.514 (0.514)** | **0.704** | **0.408** | **0.170 (0.206)** |
>
> >  *Actions taken:* We added a table that contains all used real features with a short explanation of what they represent in the appendix.
>
> | Patient Features   | Organ Features      |
> |--------------------|---------------------|
> | GENDER             | AGE_DON             |
> | DAYSWAIT_CHRON     | ALCOHOL_HEAVY_DON   |
> | ETHCAT             | BMI_DON_CALC        |
> | INIT_AGE           | COD_CAD_DON         |
> | INIT_ALBUMIN       | COLD_ISCH           |
> | INIT_ASCITES       | ETHCAT_DON          |
> | INIT_BMI_CALC      | HGT_CM_DON_CALC     |
> | INIT_BILIRUBIN     | HIST_CANCER_DON     |
> | INIT_INR           | HIST_CIG_DON        |
> | INIT_SERUM_CREAT   | GENDER_DON          |
> | INIT_SERUM_SODIUM  | NON_HRT_DON         |
> | HGT_CM_CALC        | DON_TY              |
> | WGT_KG_CALC        |                     |
> | DIAG               |                     |
>
> We hope the new empirical evidence and robustness analyses address your concerns and demonstrate that CLEXNET can meaningfully improve organ‑offer acceptance predictions in practice.
>
> Sincerely,
> Anonymous Authors

---

### Official Review · Reviewer_QPcZ · 2025-07-05

**Clarity:** 3
**Significance:** 2
**Originality:** 3
**Rating:** 4
**Confidence:** 4

**Summary:**

This paper addresses a critical limitation in organ transplantation acceptance modeling: existing machine learning models are trained on observational data and fail to generalize to counterfactual policy scenarios. To address this, the authors propose CLEXNET, a causal and explanation-guided model that integrates direction-only refusal reasons (e.g., "donor too old") as contrastive signals to guide learning. The model combines adversarial balancing to correct for policy-induced bias and a guarded explanation-guided augmentation loss to enforce alignment with refusal reasons. Empirical evaluations on synthetic data show that CLEXNET outperforms baselines in predictive accuracy, generalization, and robustness under increasing confounding. The framework is also positioned as a general recipe for counterfactual learning with direction-only supervision.

**Questions:**

1.  Do you plan to validate CLEXNET on real-world transplantation datasets, e.g., UNOS or Eurotransplant? How transferable are the findings beyond synthetic data?
2. How does CLEXNET handle potentially incorrect or missing refusal reasons? Would soft or probabilistic embeddings of reasons be more robust?
3. Have clinicians reviewed the counterfactual directions sampled from delta? Are some refusal categories more reliably directional than others?

**Ethical Concerns:**

["NO or VERY MINOR ethics concerns only"]

**Limitations:**

Yes

**Paper Formatting Concerns:**

No major formatting issues were observed. The paper appears to follow the NeurIPS 2025 formatting guidelines.

**Quality:**

3

**Strengths And Weaknesses:**

Strengths:
1) The paper tackles an important and underexplored issue in organ allocation, where offer rejections have cascading systemic effects.
2) It formalizes a new setting where only directional feedback is available (refusal codes), distinct from existing gradient- or distance-based explanation-guided learning methods.
3) The proposed CLEXNET architecture cleverly integrates domain adversarial training (to mitigate confounding) with a contrastive loss grounded in direction-only counterfactual edits.
Weaknesses:
1) All results are from synthetic datasets, which, while carefully constructed, may not fully capture the complexity and noise in real clinical data.
2) In practice, refusal codes may be noisy, inconsistently recorded, or influenced by institutional policies, which is not discussed.
3) While the mechanism is intuitive, human validation (e.g., by clinicians) on whether these augmentations reflect realistic edits is missing.

---

> ### Author Response · Authors · 2025-07-31
> **Reviewer QPcZ part 1**
>
> Dear Reviewer QPcZ,
>
> Thank you for your thoughtful and constructive evaluation of our manuscript. We appreciate your recognition of the importance of the problem, of our new “direction‑only” learning setting, and of the way CLEXNET marries adversarial debiasing with contrastive counterfactual supervision. Below we respond to each of your concerns in detail and describe the revisions that are now reflected in the updated manuscript.
>
> **Evaluation limited to synthetic data.**
> > *Actions taken:* We added semi-synthetic experiments using real‑world 2021-2024 UNOS-PTR liver‑offer data (n ≈ 1.1 M offers, 24k donors, 46k candidates).
>
> We ran additional experiments using real patients $X$ and real organs $O$ with the same synthetic setup presented in Appendix A.3. This way, the real covariate structures of patients and organs are preserved.
>
> | Model          | $\text{BCE}_\mathcal{F}$ | $\text{AUC}_\mathcal{F}$ | $\text{AUPRC}_\mathcal{F}$ | $\text{Brier}_\mathcal{F}$ |
> |----------------|------------------------:|-------------------------:|---------------------------:|---------------------------:|
> | Logistic Reg.  | 0.885 (0.672) | 0.588 | 0.771 | 0.319 (0.255) |
> | Random Forest  | 0.582 (0.275) | 0.774 | 0.891 | 0.199 (0.127) |
> | PatientNet     | 0.565 (0.963) | 0.842 | 0.920 | 0.177 (0.313) |
> | OrganITE       | 0.491 (0.828) | 0.848 | 0.922 | 0.156 (0.283) |
> | ClexNet        | **0.421 (0.662)** | **0.851** | **0.926** | **0.132 (0.234)** |
>
> Additionally, we ran experiments where every instance (consisting of a patient $X$, organ $O$, outcome $Y$ and refusal reason $R$) is real. To evaluate our models we considered the real dataset as $\mathcal{D}_{obs}$ and constructed the feasible set $\mathcal{D_F}$ by randomly rearranging real organs and real patients. We considered and encoded real refusal reasons regarding the donor age and cold ischemia time.
>
> | Model          | $\text{BCE}_\mathcal{F}$ | $\text{AUC}_\mathcal{F}$ | $\text{AUPRC}_\mathcal{F}$ | $\text{Brier}_\mathcal{F}$ |
> |----------------|------------------------:|-------------------------:|---------------------------:|---------------------------:|
> | Logistic Reg.  | 1.265 (2.536) | 0.540 | 0.279 | 0.237 (0.390) |
> | Random Forest  | 2.195 (6.829) | 0.536 | 0.268 | 0.236 (0.400) |
> | PatientNet     | 0.828 (1.348) | 0.593 | 0.308 | 0.233 (0.397) |
> | OrganITE       | 0.855 (1.402) | 0.598 | 0.308 | 0.235 (0.402) |
> | ClexNet        | **0.514 (0.514)** | **0.704** | **0.408** | **0.170 (0.206)** |
>
> >  *Actions taken:* We added a table that contains all used real features with a short explanation of what they represent in the appendix.
>
> | Patient Features   | Organ Features      |
> |--------------------|---------------------|
> | GENDER             | AGE_DON             |
> | DAYSWAIT_CHRON     | ALCOHOL_HEAVY_DON   |
> | ETHCAT             | BMI_DON_CALC        |
> | INIT_AGE           | COD_CAD_DON         |
> | INIT_ALBUMIN       | COLD_ISCH           |
> | INIT_ASCITES       | ETHCAT_DON          |
> | INIT_BMI_CALC      | HGT_CM_DON_CALC     |
> | INIT_BILIRUBIN     | HIST_CANCER_DON     |
> | INIT_INR           | HIST_CIG_DON        |
> | INIT_SERUM_CREAT   | GENDER_DON          |
> | INIT_SERUM_SODIUM  | NON_HRT_DON         |
> | HGT_CM_CALC        | DON_TY              |
> | WGT_KG_CALC        |                     |
> | DIAG               |                     |
>
> **Transferability of results.**
> Results on the semi-synthetic evaluation yield similar conclusions.
> Due to the fundamental problem of causal inference it is impossible to evaluate these models with real-world factual data as what we are testing is performance over counterfactual data generated by deviations from the current organ allocation policy. However, we would like to point out that, under our assumptions, integrating refusal reasons provided by clinicians cannot harm the performance of the model.
>
> **Potentially incorrect refusal reasons.**
> In our methodology (specifically Eq. 7) we relax Assumption 1 (which assumes that refusal reasons are correct) by adding uncertainty in the reasons through the tunable parameter $p_{ex}$. One could even further customize the model and choose different values for $p_{ex}$ based on, for example, the transplant center ID, the refusal code itself or other features. In general, we consider noisy or inaccurate refusal reasons as future work where probabilistic embeddings of reasons, both in terms of direction (accuracy) and in certainty (reliability), could offer a solution.
>
> > *Actions taken:* Added noisy and inaccurate refusal reasons to future work section.

---

> ### Author Response · Authors · 2025-07-31
> **Reviewer QPcZ part 2**
>
> **Missing refusal reasons.**
> In the UNOS-PTR dataset there is always a reason for a refusal and they are generated directly by clinicians. This is required by the current OPTN policy. We cite from section 18.3 (Recording and Reporting the Outcomes of Organ Offers) of the current OPTN policy:
>
> *"The allocating OPO and the transplant hospitals that received organ offers share responsibility for reporting the outcomes of all organ offers. ... The OPO or the OPTN must obtain PTR refusal codes directly from the physician, surgeon, or their designee involved with the potential recipient and not from other personnel."*
>
> Thus, missing reasons are not an issue in the organ setting. However, it is possible that reasons cannot be embedded into feature space (which results in the same problem) and, in other settings missing reasons could occur. In our implementation we handled this case by setting the explanatory loss $\mathcal{L}_{EXPL}$ to 0 for those instances (as the reasons are missing or cannot be embedded).
>
> > *Actions taken:* Added an additional if statement in Algorithm 1 to check whether there *is* a reason and whether it *can* be embedded.
>
> **Have clinicians reviewed the counterfactual directions.**
> We did not let clinicians review counterfactual directions. However, we would like to point out that these directions are directly provided by clinicians (we refer to the above OPTN policy citation for this). But we agree that a clinician validation of generated counterfactual samples (which depend on both the counterfactual directions and the feasible region $\mathcal{F}$) should be crucial and performed before any real world use.
>
> **Some refusal categories are more reliably directional than others.**
> Absolutely, some refusal categories are not directional or cannot be embedded in the feature space.
>
> We are grateful for your insights, which prompted substantial new experiments and refinements.
>
> Sincerely,
> Anonymous Authors

---

### Official Review · Reviewer_cg51 · 2025-07-11

**Clarity:** 3
**Significance:** 2
**Originality:** 3
**Rating:** 4
**Confidence:** 2

**Summary:**

The authors propose CLEXNET, a framework for modeling transplant offer acceptance. They note that prior approaches rely heavily on observational data and struggle to generalize to policy-relevant counterfactual scenarios. To address this, they reframe the acceptance problem as a counterfactual prediction task and leverage categorical refusal codes—recorded with each rejected organ offer—as direction-only signals for potential acceptance. Through synthetic experiments, CLEXNET demonstrates improved generalization and robustness compared to prior state-of-the-art models.

**Questions:**

* Real-world validation: Can the authors provide any empirical or qualitative evidence that the model behaves sensibly when applied to real transplant data?
* Strength of Assumption 1: Assumption 1—that "for every refusal, there exists a feasible edit along the refusal reason direction that would result in acceptance"—seems quite strong. Can the authors justify this assumption? How realistic is it in actual clinical workflows?
* Comparison to Rule-Based Systems: How does CLEXNET perform relative to existing rule-based systems or policies?

**Ethical Concerns:**

["NO or VERY MINOR ethics concerns only"]

**Final Justification:**

The authors addressed my biggest concerns, and I think they are significant. Therefore, I change my review from 3 to 4 and quality score from 2 to 3.

**Limitations:**

The lack of real world data is not discussed.

**Quality:**

3

**Strengths And Weaknesses:**

### Strengths
* The paper is well-structured, with a clear and precise formulation of the problem and a well-explained method.
* The motivation is compelling, addressing a real and important limitation in existing organ allocation models.
* The experimental results on synthetic data support the authors’ claims regarding generalization and robustness.

### Weaknesses
* The evaluation is limited to synthetic datasets; no real-world data or case studies are used to validate the model's practical utility.
* The method relies on strong assumptions about the informativeness and consistency of refusal reasons, which may not hold in real clinical settings.

---

> ### Author Response · Authors · 2025-07-31
> **Reviewer cg51 part 1**
>
> Dear Reviewer cg51,
>
> Thank you very much for your careful assessment of our submission. We appreciate your acknowledgement of the problem’s importance and of our formulation and experimental design. Below we respond point‑by‑point to every concern and describe the revisions now incorporated in the manuscript.
>
> **Evaluation limited to synthetic data.**
> > *Actions taken:* We added semi-synthetic experiments using real‑world 2021-2024 UNOS-PTR liver‑offer data (n ≈ 1.1 M offers, 24k donors, 46k candidates).
>
> We ran additional experiments using real patients $X$ and real organs $O$ with the same synthetic setup presented in Appendix A.3. This way, the real covariate structures of patients and organs are preserved.
>
> | Model          | $\text{BCE}_\mathcal{F}$ | $\text{AUC}_\mathcal{F}$ | $\text{AUPRC}_\mathcal{F}$ | $\text{Brier}_\mathcal{F}$ |
> |----------------|------------------------:|-------------------------:|---------------------------:|---------------------------:|
> | Logistic Reg.  | 0.885 (0.672) | 0.588 | 0.771 | 0.319 (0.255) |
> | Random Forest  | 0.582 (0.275) | 0.774 | 0.891 | 0.199 (0.127) |
> | PatientNet     | 0.565 (0.963) | 0.842 | 0.920 | 0.177 (0.313) |
> | OrganITE       | 0.491 (0.828) | 0.848 | 0.922 | 0.156 (0.283) |
> | ClexNet        | **0.421 (0.662)** | **0.851** | **0.926** | **0.132 (0.234)** |
>
> Additionally, we ran experiments where every instance (consisting of a patient $X$, organ $O$, outcome $Y$ and refusal reason $R$) is real. To evaluate our models we considered the real dataset as $\mathcal{D}_{obs}$ and constructed the feasible set $\mathcal{D_F}$ by randomly rearranging real organs and real patients. We considered and encoded real refusal reasons regarding the donor age and cold ischemia time.
>
> | Model          | $\text{BCE}_\mathcal{F}$ | $\text{AUC}_\mathcal{F}$ | $\text{AUPRC}_\mathcal{F}$ | $\text{Brier}_\mathcal{F}$ |
> |----------------|------------------------:|-------------------------:|---------------------------:|---------------------------:|
> | Logistic Reg.  | 1.265 (2.536) | 0.540 | 0.279 | 0.237 (0.390) |
> | Random Forest  | 2.195 (6.829) | 0.536 | 0.268 | 0.236 (0.400) |
> | PatientNet     | 0.828 (1.348) | 0.593 | 0.308 | 0.233 (0.397) |
> | OrganITE       | 0.855 (1.402) | 0.598 | 0.308 | 0.235 (0.402) |
> | ClexNet        | **0.514 (0.514)** | **0.704** | **0.408** | **0.170 (0.206)** |
>
> >  *Actions taken:* We added a table that contains all used real features with a short explanation of what they represent in the appendix.
>
> | Patient Features   | Organ Features      |
> |--------------------|---------------------|
> | GENDER             | AGE_DON             |
> | DAYSWAIT_CHRON     | ALCOHOL_HEAVY_DON   |
> | ETHCAT             | BMI_DON_CALC        |
> | INIT_AGE           | COD_CAD_DON         |
> | INIT_ALBUMIN       | COLD_ISCH           |
> | INIT_ASCITES       | ETHCAT_DON          |
> | INIT_BMI_CALC      | HGT_CM_DON_CALC     |
> | INIT_BILIRUBIN     | HIST_CANCER_DON     |
> | INIT_INR           | HIST_CIG_DON        |
> | INIT_SERUM_CREAT   | GENDER_DON          |
> | INIT_SERUM_SODIUM  | NON_HRT_DON         |
> | HGT_CM_CALC        | DON_TY              |
> | WGT_KG_CALC        |                     |
> | DIAG               |                     |
>
> **Strength of Assumption 1.**
> > *Actions taken:* We specified the conditions under which  Assumption 1 holds. Assumption 1 holds only if i) a refusal reason is given ($r \neq \emptyset$) and ii) the refusal reason can be embedded in the feature space ($\mathcal{M}(r) \neq \emptyset$).
>
> > *Actions taken:* We explicitly specified that in our methodology (specifically Eq. 7) we relax Assumption 1 by introducing the threshold acceptance probability $p_{ex}$ (instead of assuming certain acceptance like in Assumption 1).
>
> We agree that Assumption 1 is quite strong, however, it is relaxed in our methodology by a tunable parameter $p_{ex}$.
>
> In the UNOS-PTR dataset there is always a reason for a refusal and they are generated directly by clinicians. This is required by the current OPTN policy. We cite from section 18.3 (Recording and Reporting the Outcomes of Organ Offers) of the current OPTN policy:
>
> *"The allocating OPO and the transplant hospitals that received organ offers share responsibility for reporting the outcomes of all organ offers. ... The OPO or the OPTN must obtain PTR refusal codes directly from the physician, surgeon, or their designee involved with the potential recipient and not from other personnel."*
>
> Thus, it is realistic in real world clinical workflows, considering noisy or inaccurate refusal reasons as future work.
>
> > *Actions taken:* Added noisy and inaccurate refusal reasons to future work section.

---

> ### Author Response · Authors · 2025-07-31
> **Reviewer cg51 part 2**
>
> **Comparison to Rule-Based Systems.**
> Almost all existing work on organ offer acceptance is based on (probabilistic) rule-based systems/estimators. We believe that our comparison against the Random Forest benchmark shows that CLEXNET is more suited for a robust generalisation and causal estimations.
>
> We also compared a CLEXNET based organ allocation policy against *concurrent* rule-based allocation policies:
>
> | Policies                | Rank first accept           | Time to first accept (hours)          | Nonuse          |
> |-------------------------|-----------------|------------------|------------------|
> | MELD [38]                   | 9.36 (.22)      | 7.22 (.11)     | 21.8% (1.1%)   |
> | MELD‑Na [33]                | 9.66 (.32)      | 7.39 (.13)      | 22.7% (.7%)    |
> | Transplant Benefit [42]     | 9.86 (.25)      | 7.71 (.32)       | 22.5% (1%)     |
> | Acceptance-based            | **4.72 (.35)** |  **6.04 (.07)** | **18.0% (.6%)** |
>
> From these experiments we conclude that by taking acceptance into consideration it is possible to place more organs, sooner and with less offers.
>
> > *Actions taken:* Added a policy-level experiment to compare a Acceptance-based policy against other *concurrent* policies.
>
> We are grateful for your insightful feedback, which led us to strengthen the manuscript substantially. We hope the additional real‑world evaluation, relaxed assumptions, and rule‑based comparisons address your concerns about CLEXNET.
>
> Sincerely,
> Anonymous Authors
>
> ---
>
> [33] Kim, W. R., Biggins, S. W., Kremers, W. K., Wiesner, R. H., Kamath, P. S., Benson, J. T., ... & Therneau, T. M. (2008). Hyponatremia and mortality among patients on the liver-transplant waiting list. New England Journal of Medicine, 359(10), 1018-1026
>
> [38] Malinchoc, M., Kamath, P. S., Gordon, F. D., Peine, C. J., Rank, J., & Ter Borg, P. C. (2000). A model to predict poor survival in patients undergoing transjugular intrahepatic portosystemic shunts. Hepatology, 31(4), 864-871.
>
> [42] Neuberger, J., Gimson, A., Davies, M., Akyol, M., O’Grady, J., Burroughs, A., ... & Blood, U. K. (2008). Selection of patients for liver transplantation and allocation of donated livers in the UK. Gut, 57(2), 252-257.

---

> > ### Comment · Reviewer_cg51 · 2025-08-04
> > **Response to the authors**
> >
> > Dear Authors,
> >
> > Thank you for taking the time to address my concerns. The additional experiments and comparisons you provided are helpful, and I will take them into account when updating my review. I do still have reservations about Assumption 1 and its validity in real-world settings. That said, please note that my confidence rating is 2, as this topic is somewhat outside my primary area of expertise.
> >
> > Sincerely,
> > Reviewer

---

> > > ### Author Response · Authors · 2025-08-06
> > > **Follow up cg51**
> > >
> > > Dear Reviewer cg51,
> > >
> > > Thank you very much for your prompt follow-up and for weighing our additional experiments and analyses. We truly appreciate your engagement with our submission.
> > >
> > > We actually agree with your stance on Assumption 1 and decided to empirically test support over real data for this property.
> > >
> > > > *Actions taken:* Added an empirical support study for Assumption 1 over real UNOS-PTR data (n ≈ 1.1 M offers).
> > >
> > > For each negative instance ($(X,O)$-pair) with a refusal reason, we tried to match it with a positive instance that i) satisfies that reason and ii) is within a certain range of the negative instance. Here are the results:
> > >
> > > | Allowed matching range (Euclidean distance) | Support for donor age related reasons  | Support for cold ischemic time related reasons  | Overall Support |
> > > |--------|------:|------:|--------:|
> > > | $\leq1$      | 0.1%   | 0.1%   | 0.1%    |
> > > | $\leq2$      | 7.0%   | 4.7%   | 6.2%    |
> > > | $\leq3$      | 71.7%  | 68.8%  | 70.7%   |
> > > | $\leq4$      | 98.7%  | 97.8%  | 98.4%   |
> > > | $\leq5$      | 99.9%  | 99.8%  | 99.9%   |
> > > | $\leq6$      | 100.0% | 100.0% | 100.0%  |
> > >
> > > If we allow for matching negatives with positives within a Euclidean distance of 3 (consider that the feature space has 76 dimensions after OHE), we can find suitable positives that satisfy Assumption 1 for over 70% of the negatives. If we increase the distance to 5, then Assumption 1 is almost always satisfied.
> > >
> > > While these findings show support for Assumption 1 over the *observed data* we recognise that Assumption 1 may not always hold in practice. We have therefore added a dedicated discussion section to make its limitations transparent to readers in which we have added a short blueprint on relaxed and probabilistic variants, for example by using an instance-level $p_{ex}$. This can be used for instance if a transplant center consistently generates unreliable reasons.
> > >
> > > We hope these new analyses and the expanded discussion clarify both the empirical support for Assumption 1 and the practical safeguards we have introduced to mitigate its inevitable imperfections. As always, we are grateful for your careful feedback and would be happy to elaborate on any remaining questions.
> > >
> > > Sincerely,
> > > The Authors

---

> > > > ### Comment · Reviewer_cg51 · 2025-08-08
> > > > **Re: Follow up**
> > > >
> > > > Dear authors,
> > > >
> > > > Thank you for providing additional analysis. I believe it is a valuable addition to the paper. I think it ultimately clarifies my concerns.
> > > >
> > > > Sincerely, Reviewer

---

### Note · Authors · 2025-08-12

Dear AC and Reviewers,

Thank you for the thoughtful interaction throughout the review process. We appreciate the time and effort invested in reviewing our work and providing valuable feedback.

We are pleased to have been able to address the concerns raised, and believe this has ultimately strengthened and improved the quality of our paper.

Kind regards,
Anonymous Authors

---

### Decision · Program_Chairs · 2025-09-17

**Decision:**

Accept (poster)

**Comment:**

(a) Scientific claims and findings
The paper introduces CLEXNET, a framework for modeling transplant offer acceptance. Unlike prior models that rely solely on observational data, CLEXNET reframes the task as counterfactual prediction and leverages categorical refusal codes as direction-only signals. Synthetic experiments show improved generalization and robustness compared to prior approaches.

(b) Strengths

Compelling motivation addressing an important problem in organ allocation.

Clear formulation and well-presented method.

Novel use of refusal codes as direction-only counterfactual signals.

Innovative integration of adversarial training with a contrastive explanation loss.

Demonstrates improved robustness and generalization in synthetic experiments.

(c) Weaknesses

Initial results limited to synthetic datasets, raising questions of real-world applicability.

Assumptions about the reliability and informativeness of refusal codes may not hold in practice.

Lack of human/clinical validation.

Unclear delineation of which parts of the method are novel.

(d) Reasons for decision
I recommend acceptance. The paper presents a novel and well-motivated framework addressing an important, underexplored issue in organ allocation. Despite initial concerns about evaluation on only synthetic data, the rebuttal addressed this by providing results beyond synthetic datasets, which reassured reviewers. The theoretical and methodological contributions, together with the relevance of the problem, justify acceptance.

(e) Rebuttal and discussion
During review, concerns centered on the reliance on synthetic datasets, unclear novelty of some components, and assumptions about refusal code quality. In rebuttal, the authors presented additional results beyond synthetic data, alleviating the primary concern. Reviewers were satisfied with these clarifications, and consensus shifted positively, leading to the final recommendation of acceptance.